# A noncoding variant confers pancreatic differentiation defect and contributes to diabetes susceptibility by recruiting RXRA

Yinglei Li[1], Ran Zheng[1], Lai Jiang[1], Chenchao Yan[1], Ran Liu[1], Luyi Chen[1], Wenwen Jin[1], Yuanyuan Luo[1], Xiafei Zhang[1], Jun Tang[2], Zhe Dai[2] & Wei Jiang ●[1,3,4] ✉

Human genetics analysis has identified many noncoding SNPs associated with diabetic traits, but whether and how these variants contribute to diabetes is largely unknown. Here, we focus on a noncoding variant, rs6048205, and report that the risk-G variant impairs the generation of PDX1+/NKX6-1+ pancreatic progenitor cells and further results in the abnormal decrease of functional β cells during pancreatic differentiation. Mechanistically, this risk-G variant greatly enhances RXRA binding and over-activates *FOXA2* transcription, specifically in the pancreatic progenitor stage, which in turn represses NKX6-1 expression. Consistently, inducible FOXA2 overexpression could phenocopy the differentiation defect. More importantly, mice carrying risk-G exhibit abnormal pancreatic islet architecture and are more sensitive to streptozotocin or a high-fat diet to develop into diabetes eventually. This study not only identifies a causal noncoding variant in diabetes susceptibility but also dissects the underlying gain-of-function mechanism by recruiting stage-specific factors.

Diabetes mellitus is a complicated metabolism syndrome characterized by hyperglycemia, mainly resulting from loss or dysfunction of pancreatic β cells. Both genetic predisposition and environmental exposure contribute to the pathogenesis of diabetes. Scientists have revealed that single gene mutation could cause certain types of diabetes, such as neonatal diabetes, maturity-onset diabetes of the young (MODY), and syndromic diabetes[1]. In monogenic diabetes, most mutated genes are transcription factors essential in β cell development or insulin secretion[2]. For example, MODY3, known as HNF1A, is expressed during the development of endodermal organs, and diabetic patients with MODY3 mutation exhibit a defect in glucose sensing in the β cells[3]; NEUROG3 is crucial for the pancreatic endocrine formation, and biallelic *NEUROG3* mutations could cause permanent neonatal diabetes with lower C-PEP level and undetectable GLUCAGON level[4]. These monogenic mutations are identified mainly through whole-exome sequencing surveying coding regions[1,5,6]. In addition, genome-wide association studies (GWAS) have identified numerous single-nucleotide polymorphism (SNP) variants associated with diabetes-related traits, such as fasting glucose (FG) level and β cell function. However, 88% of disease-associated SNPs are located in noncoding intronic (45%) or intergenic (43%) regions[7], and most of these SNPs are annotated solely based on statistical association. Whether these noncoding variants are functional and how they contribute to the emerging incidence of disease must be studied[8].

Studies have increasingly acknowledged that visible species-specific differences exist in gestation period, morphology, and the spatial and temporal regulation of gene expression during embryonic development and organ formation between humans and mice[9,10]. For

[1]Department of Biological Repositories, Frontier Science Center for Immunology and Metabolism, Medical Research Institute, Zhongnan Hospital of Wuhan University, Wuhan University, Wuhan 430071, China. [2]Department of Endocrinology, Zhongnan Hospital of Wuhan University, Wuhan, China. [3]Human Genetics Resource Preservation Center of Wuhan University, Wuhan, China. [4]Hubei Provincial Key Laboratory of Developmentally Originated Disease, Wuhan, China. ✉e-mail: jiangw.mri@whu.edu.cn

example, heterozygous inactivating mutations in *GATA6* were shown to impair human pancreatic lineage commitment[11,12], and it has been observed in more than half of human patients with pancreatic agenesis[13,14], which could not be expected from mouse study that *Gata6* heterozygous null mice show no obvious phenotype[15]. In general, increasingly accumulated differences between human and mouse β cell development, such as cell-cell interaction and islet structure, have prevented direct translation of the knowledge learned in mouse study to human[16]. However, human embryonic stem cells (ESCs) and induced pluripotent stem cells (iPSCs), collectively known as pluripotent stem cells (PSCs), provide an alternative approach to dissecting the developmental mechanisms and potential pathological role of mutations identified from clinical studies[17]. Indeed, significant achievements have been made to obtain stem cell-derived pancreatic β cells capable of responding to glucose stimulation and rescuing hyperglycemia in diabetic mouse models[18–21].

Using the in vitro pancreatic lineage differentiation system, scientists can dissect the underlying mechanisms of mutations for monogenic diabetes. For instance, a patient-derived iPSC line carrying one allele of *HNF1B*[S148L] mutation, which could cause an ectopic activation of certain critical pancreatic transcription factors such as *PDX1*, *PTF1A*, *GATA4*, and *GATA6*, exhibits a severe defect in pancreatic development[22]. Specification of β cells from *HNF1A*[R272C] mutated iPSCs is unperturbed; however, this mutation leads to the deregulation of membrane potential-related genes causing insulin hypersecretion[23], while *HNF1A*[R200Q] and *HNF1A*[P291fsinC] impair basal and glucose-stimulated insulin secretion[24,25]. Human ESCs carrying the patient variant *ONECUT1*[E231X] fail to transactivate *NKX6-1* and *NKX2.2* enhancers, reducing PDX1+/NKX6-1+ progenitor cells[26]. In addition, human iPSCs carrying two *PAX4* risk alleles tend to generate increased polyhormonal endocrine cells, which exhibit reduced insulin content[27]. More interestingly, the iPSCs carrying *PDX1* − 18C>T variant in the 5′-untranslated region have attenuated probability to differentiate into pancreatic progenitors[28]. These remarkable works demonstrate the power of human PSC models to reflect various aspects of the pathophysiology of monogenic diabetes and offer an alternative platform to dissect the upstream and downstream mechanisms underlying the genetic variations.

Here, we focus on a noncoding SNP rs6048205 (NC_000020.11:22578963:A>G), located downstream of the transcription termination site of *FOXA2* gene with 2 kilo base-pairs away (Fig. 1a). This variant is associated with the symptoms of higher fasting-glucose level and impaired β cell function[7,29,30]. According to the statistics of Allele Frequency Aggregator (ALFA) project[31], global frequency of minor allele G frequency of rs6048205 is approximately 5.8%, with a higher frequency in African (approximately 17.6%) and East Asian population (approximately 19%) (Supplementary Fig. 1a). To address the function of this variant in pancreatic differentiation and its potential pathological role, we first establish the isogenic pluripotent stem cell lines carrying different variant (A or G) and analyze the differentiation phenotype. By motif analysis and biochemical assay, we reveal that RXRA could specifically bind to the risk allele and activate the transcription of *FOXA2* gene, thus repressing the downstream *NKX6-1* expression. Finally, we construct a mouse model carrying the risk allele and observe abnormal glycemia upon diabetic stimuli, including STZ and high-fat diet, illustrating the pathological role of this noncoding variant for diabetes susceptibility.

## Results
### rs6048205-G results in defective pancreatic progenitor formation
To determine the role of the rs6048205 variant in pancreatic development, we utilize the directed differentiation of human PSCs to pancreatic β-like cells. First, we constructed the isogenic cell lines carrying different genotypes including wild-type (A/A), heterozygote (A/G), homozygote with risk alleles (G/G) in human iPSC line PGP1 via CRISPR/Cas9-mediated knock-in technology (Supplementary Fig. 1b). Sanger sequencing analysis of the potential off-target sites of the sgRNA used to target the SNP rs6048205 locus did not reveal any evidence of off-target edits (Supplementary Fig. 1c). Additionally, all cell lines exhibited a normal karyotype as determined by G-band karyotype analysis (Supplementary Fig. 1d). To further exclude on-target effects[32], we conducted genomic qPCR to determine the copy number of alleles at the SNP site and found no differences (Supplementary Fig. 1e). Meanwhile, there were no significant differences in gene expression of pluripotent markers observed (Supplementary Fig. 1f), and these cell lines were then subjected for stepwise pancreatic β-like cell differentiation according to the well-established protocols[20,21,33].

We mainly checked the four stages by the respective vital markers: SOX17 for definitive endoderm (DE), PDX1 for pancreatic progenitor 1 (PP1) stage, PDX1 and NKX6-1 for pancreatic progenitor 2 (PP2) stage, INSULIN/C-PEP combined with PDX1/NKX6-1 for the β-like cell stage (Fig. 1b, c). As the data shows, the PSCs with risk allele G could normally differentiate into DE with high efficiency, assayed by the percentage of SOX17-positive cells using flow cytometry (Supplementary Fig. 2a), and further supported by the immunostaining results (Supplementary Fig. 2b) and quantitative reverse-transcription PCR (RT-qPCR) data (Supplementary Fig. 2c). Those results indicated that the risk allele G did not affect DE differentiation from human PSCs. In PP1 stage, the percentage of PDX1-positive progenitors in cells carrying allele G was comparable with allele A evaluated by immunofluorescence (Supplementary Fig. 2d) and flow cytometry (Fig. 1d and Supplementary Fig. 2e). Besides, allele G did not affect *PDX1* expression, but the expression of *NKX6-1* was dramatically decreased (Fig. 1e).

As *NKX6-1* reached the expression peak at PP2 stage (Fig. 1c), we further evaluated the following PP2 stage. Immunofluorescence results showed no significant difference in the percentage of PDX1-positive cells again (Fig. 1f, g), albeit accompanied by slightly but statistically significant lower *PDX1* and *PTF1A* expression in allele G group (Fig. 1h). Surprisingly, consistent with the dramatically decreased *NKX6-1* mRNA level (Fig. 1h), the immunofluorescence (Fig. 1f, g) and flow cytometry data (Fig. 1i, j) indicated that the percentage of PDX1 and NKX6-1 double-positive cells was remarkably decreased in allele G group.

Additionally, we derived A/- and G/- cells from the ESC line HUES8, which is heterozygous in the rs6048205 locus (Supplementary Fig. 3a). These new lines with a distinct genetic background exhibited similar phenotypes consistent with those in A/A and G/G cells. Notably, the G/-genotype did not affect the expression levels of key transcription factors during the definitive endoderm (DE) stage (Supplementary Fig. 3b). Moreover, G/- did not affect *PDX1* expression levels or the percentage of PDX1-positive cells in the PP stage. However, it significantly reduced the percentage and expression levels of NKX6-1-positive cells (Supplementary Fig. 3c–f). These findings support our previous results, reinforcing the robustness of our conclusions regarding the defects in pancreatic progenitor cell differentiation caused by the risk variant G.

To comprehensively understand the impact of the risk allele G on pancreatic progenitor development, we performed RNA-seq on PP2 stage. With the cutoff of fold-change >1.5 and $p < 0.05$, 1116 differentially expressed genes (DEGs) were identified between allele G and allele A, with 645 down-regulated and 471 up-regulated (Fig. 1k). Consistent with the previous results, *NKX6-1* was remarkably reduced together with other pancreatic key transcription factors such as *SOX9* and *PDX1* (Fig. 1k) on PP2 stage. Similarly, gene ontology (GO) analysis showed that the significantly down-regulated genes in allele G enriched terms about pancreatic development, endocrine system development, and export from cell (Fig. 1l). In contrast, up-regulated

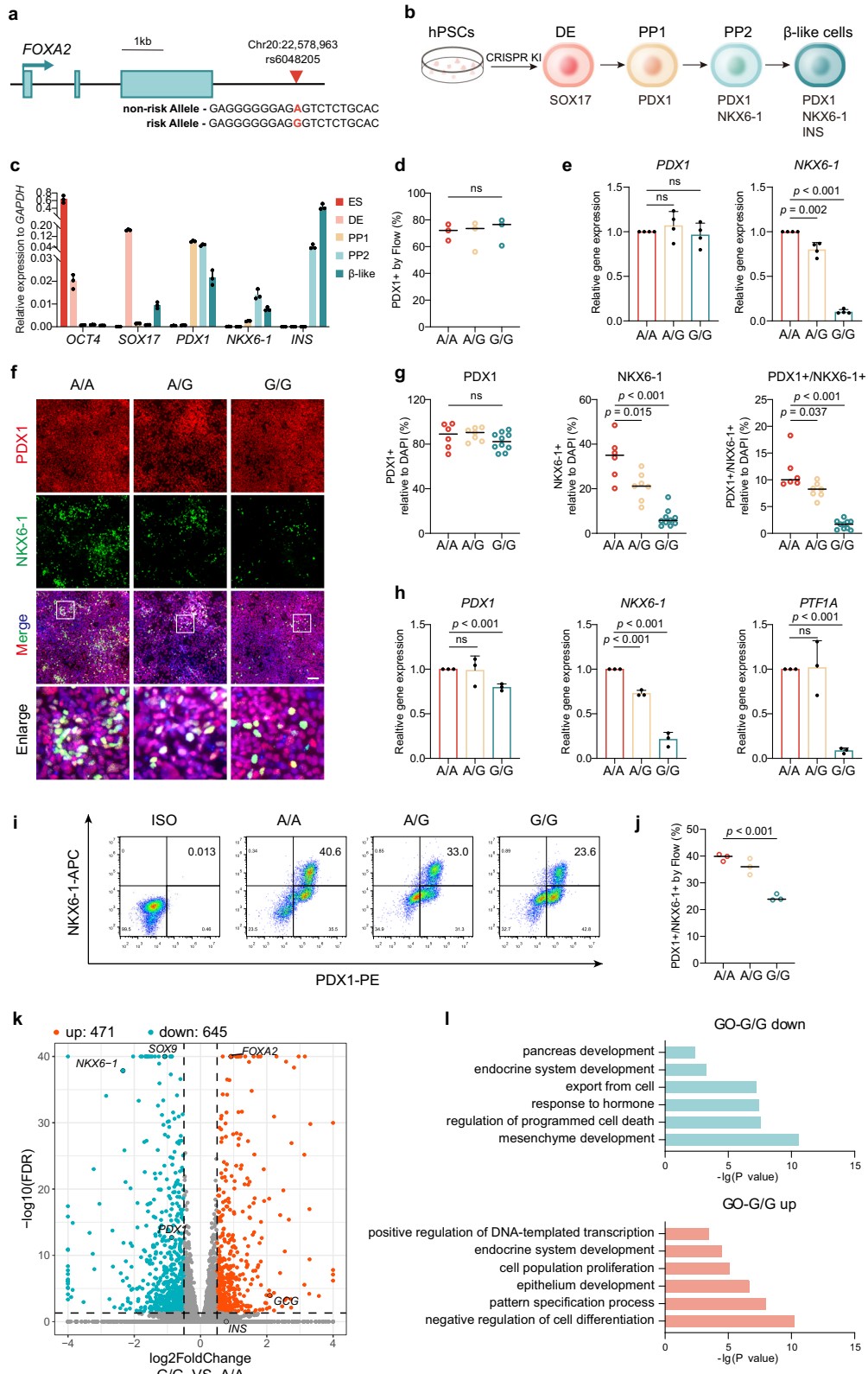

genes enriched terms in negative regulation of cell differentiation, epithelium development (Fig. 1l). The data provided strong support for the notion that the rs6048205-G allele indeed caused a reduction in the generation of NKX6-1+/PDX1+ pancreatic progenitors, resulting in defective pancreatic differentiation.

## rs6048205-G leads to a reduction of functional β cells at the β-like stage

We next wondered whether allele G would influence the following differentiation step of pancreatic β-like cells. We first examined several critical markers' mRNA and protein expression levels, including *PDX1*,

**Fig. 1 | rs6048205-G affected pancreatic progenitor cell differentiation.**
**a** Schematic diagram of the SNP rs6048205 location. Locus of rs6048205 was shown in red. **b** Schematic of pancreatic differentiation protocol from human PSCs. Differentiation stages and key markers are shown. hPSCs, human pluripotent stem cells; DE, definitive endoderm; PP1, pancreatic progenitors 1 marked by PDX1; PP2, pancreatic progenitors 2 marked by PDX1 and NKX6-1; β-like cells, insulin-producing β-like cells. **c** Time-course mRNA levels of key stage-specific markers detected by RT-qPCR ($n = 3$ independent experiments). **d** Statistical analysis of PDX1-positive pancreatic progenitor cells in PP1 detected by flow cytometry in Supplementary Fig. 2d ($n = 3$ independent experiments). **e** mRNA levels of PP1 markers (*PDX1* and *NKX6-1*) quantified by RT-qPCR. Gene expression levels were normalized to *GAPDH* and then further normalized to the A/A group ($n = 4$ independent experiments). **f, g** Representative immunofluorescence images (**f**) of PP2 cells stained by PDX1 (red) and NKX6-1 (green) and quantitative analysis for the percentage of cells expressing each marker (**g**) ($n = 6$ images in A/A, $n = 7$ in A/G and

$n = 10$ in G/G). Nuclei were counterstained with DAPI (blue). Scale bar, 100 μm. **h** mRNA levels of PP2 markers (*PDX1*, *NKX6-1* and *PTF1A*) detected by RT-qPCR. Gene expression levels were normalized to *GAPDH* and then further normalized to the A/A group ($n = 3$ independent experiments). **i, j** Flow cytometric analysis of PDX1- and NKX6-1-positive cells in PP2 stage (**i**) and quantitative analysis (**j**) ($n = 3$ independent experiments). **k** Scatterplot showing differentially expressed genes (DEGs) identified by RNA-seq of G group and A group in PP2 stage ($n = 2$ independent samples). Up-regulated and downregulated genes in the allele G group were shown in red and blue, respectively. **l** GO analysis of the up- and down-regulated genes in the risk allele G group compared to the A group. Data are presented as the mean ± SD. Statistical significance was determined using the unpaired, two-tailed *t*-test in (**d, e, g, f, j**) (ns not significant), one-sided hypergeometric test with multiple comparison adjustments was used in (**l**). Source data are provided as a Source Data file.

*NKX6-1, INS, GCG* and *SST*. Interestingly, the mRNA levels of *INS, GCG* and *SST* were increased in allele G group (Fig. 2a), together with the increased ratio of C-PEP-positive β-like cells (Fig. 2d, f); the mRNA expression of *NKX6-1* in β-like cell stage for allele G group was caught up (Fig. 2a) and indeed the percentage of NKX6-1 in allele G group was dramatically boosted to comparable level to allele A group (Fig. 2b, c). However, most NKX6-1-positive cells did not co-stain with PDX1 (Fig. 2b, c). Most C-PEP-positive cells in the G allele group were also positive for GLUCAGON (Fig. 2d, f) as well as SOMASTATIN (Fig. 2e, f), indicating the presence of polyhormonal cells, a type of unmatured endocrinal cells[34–37]. This polyhormonal observation was confirmed by flow cytometry analysis (Supplementary Fig. 4b, c). Glucose-stimulated insulin secretion (GSIS) assays showed that the mutant cells exhibited reduced Insulin and C-Peptide secretion in response to high glucose stimulation (Fig. 2g, h).

Interestingly, the monohormonal GCG and SST populations significantly increased, suggesting that the risk variant G can hinder β cell development, leading to a shift toward other endocrine lineages (Supplementary Fig. 4a). Furthermore, overall apoptosis rate was very low (< 2%) during differentiation and no apparent changes in apoptosis were observed in the G allele variant cells compared to controls at DE, PP1 or PP2 stages (Supplementary Fig. 4d–g). These data indicate that the rs6048205-G risk allele impairs β cell differentiation and function.

### rs6048205-G functions through FOXA2

Next, to understand the downstream target of rs6048205-G to affect pancreatic differentiation (inhibiting the formation of NKX6-1-positive cells at PP2 stage), we inquired about the expression levels of genes located within 2 Mb away from the rs6048205 locus. By surveying the RNA-seq data, we found *FOXA2* exhibited the most dramatic change between allele G and allele A groups (Fig. 3a). We also confirmed that the *FOXA2* expression was robustly elevated in PP stage by RT-qPCR (Fig. 3b, Supplementary Fig. 3b, c, e) and Western blot (Fig. 3c). In addition, analyzing the published FOXA2 chromatin immunoprecipitation followed by sequencing (ChIP-seq) data of pancreatic progenitor cells[38], we found more than half of the differentially expressed genes (602/1116) between allele G and allele A at PP2 stage were FOXA2-binding genes (Supplementary Fig. 5a). These data suggested that the defected pancreatic differentiation due to rs6048205-G may be caused by abnormal activation of *FOXA2*.

FOXA2 is identified as a pioneer transcription factor in pancreatic development and thus required for normal pancreatic differentiation[38]. However, whether FOXA2 overexpression could result in defective pancreatic differentiation is unknown. To test this hypothesis, we introduced a doxycycline-induced FOXA2 together with mCherry cassette at endogenous *AAVS* locus of human ESC line HUES8[39,40] (Fig. 3d). Doxycycline could indeed activate mCherry expression together with FOXA2 in ESCs (Supplementary Fig. 5b, c). Next, we confirmed that the addition of doxycycline in PP stage during

differentiation was able to overexpress FOXA2 in both mRNA (Supplementary Fig. 5d) and protein levels during differentiation in this line (Fig. 3e). Interestingly, the mRNA expression of *NKX6-1* was dramatically decreased upon FOXA2 overexpression in pancreatic progenitors. At the same time, *PDX1* showed mild reduction (Supplementary Fig. 5d). This notion was supported by the Western blot and immunostaining results that NKX6-1 protein was dramatically reduced but PDX1 remained comparable in FOXA2-overexpressed progenitors (Fig. 3e–g and Supplementary Fig. 5e).

Next, we examined the generation of β-like cells during the differentiation with FOXA2 overexpression in PP stage. The results showed that FOXA2 overexpression contributed to defective pancreatic β cell differentiation, mainly reflected by the decreased mRNA levels of pancreatic marker genes (Supplementary Fig. 5f) as well as the percentage of PDX1-, NKX6-1- and C-PEP-positive cells (Supplementary Fig. 5g).

To further understand the effect of the induced FOXA2 overexpression in human pancreatic progenitor cells, we performed RNA-seq experiment using the FOXA2-overexpressing and control PP2 samples. With the cutoff fold-change > 1.5 and *p* < 0.05, we identified 4468 DEGs, of which 2260 were up-regulated and 1808 were down-regulated (Fig. 3h). The mRNA levels of *FOXA2, PDX1* and *NKX6-1* showed the similar trend to the RT-qPCR results (Supplementary Fig. 5d). Consistently, genes downregulated in FOXA2-overexpressed PP2 cells were enriched in terms including pancreatic development, insulin secretion and peptide hormone secretion, and the genes up-regulated involved in regulation of DNA-binding transcription factor activity, epithelial cell differentiation, and cell population proliferation (Fig. 3i). More importantly, half of the DEGs in allele G compared with allele A (576/1116) were also changed upon FOXA2 overexpression, and these overlapped genes enriched in terms including endocrine pancreas development, export from cell, and MODY-related genes (Fig. 3j). Notably, 59% of these overlapped DEGs (342/576) were FOXA2 binding genes, while about 42% (1888/4468) of DEGs due to FOXA2 overexpression were FOXA2 binding genes (Supplementary Fig. 5h). These data together indicated that rs6048205-G impaired pancreatic differentiation possibly through higher expression of FOXA2, and FOXA2 overexpression in progenitor stage could indeed phenocopy the rs6048205-G in molecular level.

Since NKX6-1 decreased dramatically in both the allele G group and the FOXA2-overexpressed PP2 stage, we were next wondering whether NKX6-1 was the direct target of FOXA2. We first checked the ChIP-seq data of FOXA2 in pancreatic differentiation[38] and observed that FOXA2 could be recruited to *NKX6-1* locus (Fig. 3k) which was validated in our PP2 samples (Fig. 3l). To investigate whether the FOXA2-binding regions from *NKX6-1* locus were indeed functionally responsible to FOXA2 protein, we cloned these sequences to luciferase vector and then co-transfected with FOXA2 cDNA. The results showed that the overexpression of FOXA2 led to significantly repressed *NKX6-1*

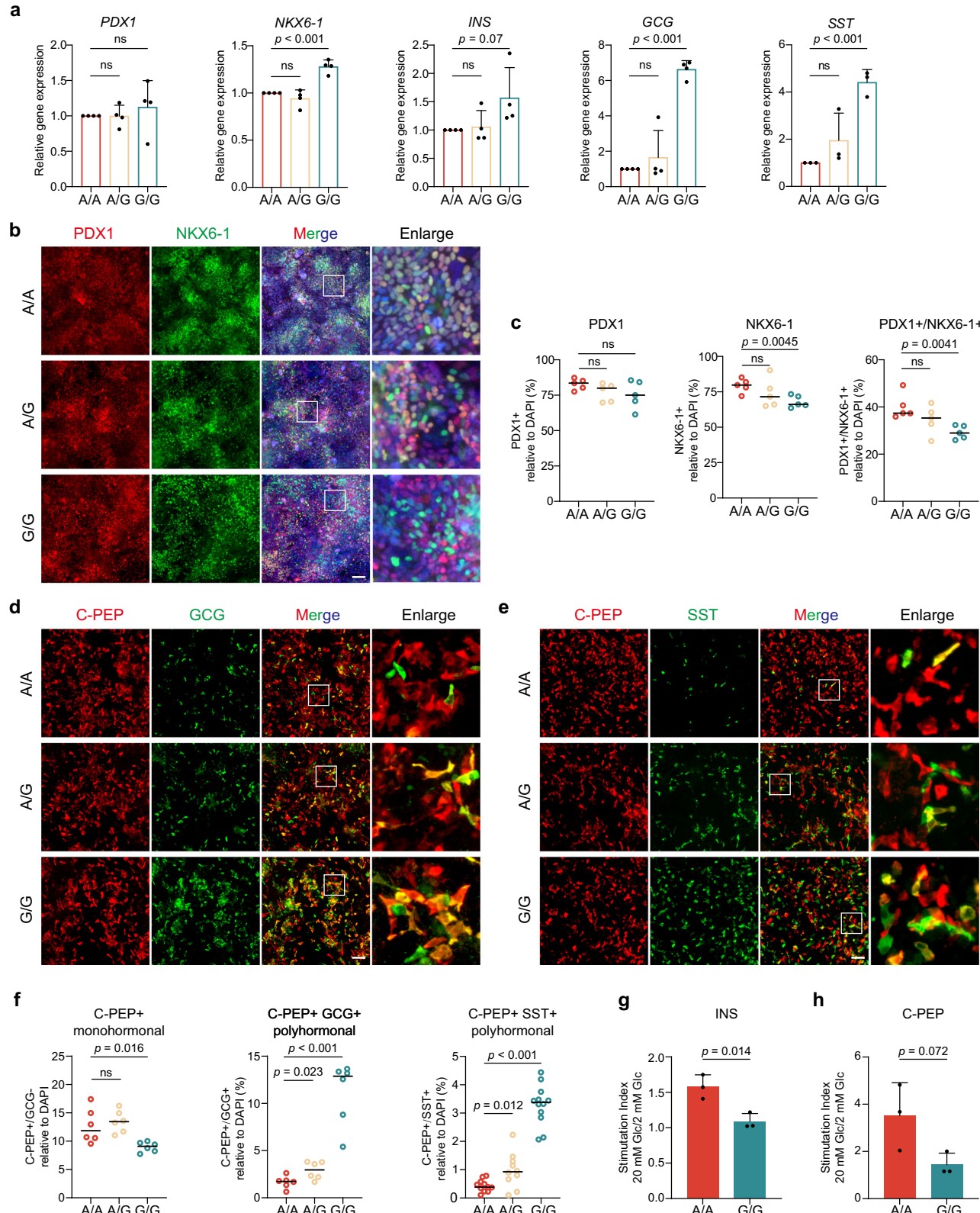

luciferase activity at the transcription start site (Fig. 3m). Consistent with this data, 36% of the SNP-affected genes (409/1116) were NKX6-1 binding genes determined by ChIP-seq analysis (GSE131817)[26]. In comparison, 32% of FOXA2-overexpression affected genes (1441/4468) were NKX6-1 binding genes (Supplementary Fig. 5i). These results collectively supported that FOXA2 could bind to *NKX6-1* locus and repressed the *NKX6-1* transcription.

**RXRA activates FOXA2 expression through rs6048205-G**

As risk allele G contributed to the over-activation of FOXA2, we supposed that rs6048205 may located in the regulatory region of *FOXA2*. We examined the chromatin environment, including accessibility and histone modifications at the gene locus of SNP rs6048205 based on the published datasets (GSE148368, GSE104840). We found rs6048205 is located near an open chromatin region and exhibits H3K4me1 and

**Fig. 2 | rs6048205-G led to a reduction of functional β cells at the β-like cell stage. a** mRNA levels of β cell markers (*PDX1, NKX6-1, INS, GCG,* and *SST*) detected by RT-qPCR at the β-like cell stage. Gene expression levels were normalized to *GAPDH* and then further normalized to the A/A group (*n* = 4 independent experiments). **b, c** Representative immunofluorescence images of β-like cell stage stained by PDX1 (red) and NKX6-1 (green) (**b**) and quantitative analysis for the percentage of cells expressing each marker (**c**) (*n* = 5 images). Scale bar, 100 μm. **d** Representative immunofluorescence images of β-like cell stage stained by C-PEP (red) and GCG (green). Scale bar, 100 μm. **e** Representative immunofluorescence images of β-like cell stage stained by C-PEP (red) and SST (green). Scale bar, 100 μm.

**f** Quantitative analysis for the percentage of C-PEP+ monohormonal cells (*n* = 6 images), C-PEP+/GCG+ polyhormonal cells (*n* = 6 images) and C-PEP+/SST+ polyhormonal cells (*n* = 11 images in A/A, *n* = 10 in A/G and *n* = 12 in G/G) in Fig. 2d, e. **g, h** Glucose-stimulated insulin (**g**) and C-PEP (**h**) secretion assay comparing the function of G/G and A/A β-like cells. The stimulation index is calculated as the ratio of C-peptide released at high glucose concentration (20 mM) to low glucose (2 mM) (*n* = 3 independent experiments). Data are presented as the mean ± SD. Statistical significance was determined using the unpaired, two-tailed *t*-test in (**a, c, f, g, h**) (ns not significant). Source data are provided as a Source Data file.

H3K27ac modifications in differentiated pancreatic progenitor cells, suggesting that SNP rs6048205 is within an enhancer-like regulatory region (Supplementary Fig. 6a).

To confirm regulatory function of SNP rs6048205, we first conducted luciferase reporter assays by cloning the DNA sequence containing the SNP. We assayed in PANC1, which harbors pancreatic lineage context. The results showed that, compared with allele A, allele G exhibited robust transcription activity in PANC1 cells (Fig. 4a). This effect was not observed in HEK-293T cells, an embryonic kidney cell type (Supplementary Fig. 6b), suggesting that rs6048205 plays a vital role in a cell type-specific manner. Since rs6048205 enhances FOXA2 expression specifically in PP stage, we further performed this luciferase assay in DE, PP1, and PP2 samples. The results demonstrated that this regulation occurred only in pancreatic progenitor samples (including PP1 and PP2 stages) but not in DE samples (Fig. 4a), indicating that rs6048205 functions specifically in the pancreatic progenitor cell stage.

To determine which transcription factor likely contributed to such interesting stage-specific regulation, we predicted the potential binding motif within a 20 bp region surrounding rs6048205 by PROMO[41,42]. Interestingly, A > G mutation seemed to create a new binding motif for RXRA (Supplementary Fig. 6c). JASPAR analysis also showed that RXRA potentially binds to allele G rather than allele A (Fig. 4b), which together suggested RXRA might be the critical factor utilizing the variant G.

Therefore, we first confirmed that the oligo containing rs6048205-G, but not the allele A, could indeed directly bind to RXRA protein by electrophoretic mobility shift assay (EMSA) (Fig. 4c). Next, we transiently overexpressed RXRA in 293T cells with the luciferase system which harbored the regulatory sequence containing allele A or allele G. The result showed that RXRA exhibited significantly higher transcription activation capacity on allele G than on allele A (Fig. 4d). Then we performed a ChIP assay followed by the allele-specific qPCR[43,44] in Huh7 cell line which is heterozygous in rs6048205 (A/G). Excitingly, we observed that RXRA preferentially bound to the allele G compared with allele A (Fig. 4e). We further determined the RXRA binding efficiency in pancreatic progenitor cells derived from different human PSCs carrying allele G or allele A by ChIP-qPCR. Similarly, the cells with allele G exhibited higher RXRA bound compared with the cells with allele A (Supplementary Fig. 6d).

We checked the RXRA expression during pancreatic differentiation, and we found RXRA was highly expressed from DE to PP stages (Fig. 4f). However, immunostaining data showed that RXRA was mainly localized in the cytoplasm at DE stage and then exhibited nucleus-localization at PP stage (Fig. 4g), providing a potential explanation on the stage-specific activation of FOXA2 co-mediated by allele G and RXRA.

Additionally, to confirm the regulatory role of RXRA on FOXA2 expression, we applied two different RXR agonists, bexarotene (Bexa) and LG100268 (LG268) in PP stage derived from PSCs carrying the heterozygote genotype on rs6048205 locus. The RT-qPCR results showed that the expression levels of RXR target genes were increased along with Bexa and LG268 treatment in a dose-dependent manner

(Supplementary Fig. 6e), indicating the efficacy of RXR activation by these chemicals. Consistent with the previous notion, RXR agonists elevated FOXA2 expression in protein and mRNA levels under the highest dosage treatment (Fig. 4h, i). Moreover, both the percentage of NKX6-1-positive cells and the expression of *NKX6-1* were dramatically decreased upon RXRA activation. In contrast, *PDX1* expression was unchanged (Fig. 4h–j, Supplementary Fig. 6f). Notably, RXR activation by the Bexa and LG268 did not induce cell apoptosis (Supplementary Fig. 6g).

Furthermore, RXR inhibitor (UVI3003) treatment during pancreatic progenitor differentiation of G/G cells did not affect *PDX1* expression (Supplementary Fig. 6h) but resulted in a reduction in *FOXA2* levels and a slight increase in *NKX6-1* expression (Fig. 4k). Conversely, in A/A cells, *FOXA2* expression was not reduced, and *NKX6-1* expression showed no significant variation (Supplementary Fig. 6i). Those findings further indicated that rs6048205-G regulated FOXA2 expression in a RXRA-dependent manner.

## Mice with rs6048205-G exhibit more sensitivity to STZ or HFD treatment

Notably, we found that the sequence around rs6048205 was conserved between mice and humans (Supplementary Fig. 7a), which provided an opportunity to evaluate the in vivo function of the rs6048205 variant. We therefore introduced the point mutation A > G in wild-type mice to test the potential physiological effect in glucose metabolism and islet function (Fig. 5a). No differences in body weight from 2 weeks to 10 weeks were observed in those knock-in mice compared with their age-matched controls (Supplementary Fig. 7b). All mice had comparable random blood glucose levels (Supplementary Fig. 7c). However, the fasting blood glucose level was slightly but significantly higher in allele G mice than in wild-type allele A controls (Fig. 5b).

An intraperitoneal glucose tolerance test (ipGTT) showed mild but significantly increased glucose intolerance in allele G mutants compared to WT mice (Fig. 5c), while no difference was observed in intraperitoneal insulin tolerance test (ipITT) measuring the sensitivity to exogenously administered insulin (Supplementary Fig. 7d). We further examined the islet structure of mice with different genotypes. The immunostaining data of adult pancreas showed that allele G mice had decreased insulin-positive β cells and increased glucagon-positive α cells (Fig. 5d, e). This prompted us to examine the structure of the embryonic pancreas to investigate the impact of rs6048205 on pancreatic development. We found that NKX6-1 positive cells were markedly reduced in the E13.5 embryonic pancreas (Fig. 5f, g). Additionally, in the pancreas of E18.5 mice, we observed a low proportion of polyhormonal cells in the embryonic mouse pancreas. However, quantifying these polyhormonal cells in the G/G group revealed a significant increase (Supplementary Fig. 7e, f) consistent with the phenotype observed in the PSC differentiation assay.

The fasting blood glucose, ipGTT, and islet structure results indicated certain impairment of pancreatic β cell function, and this prompted us to challenge these mice under pathological conditions such as streptozotocin (STZ) or high-fat diet (HFD). We first applied a

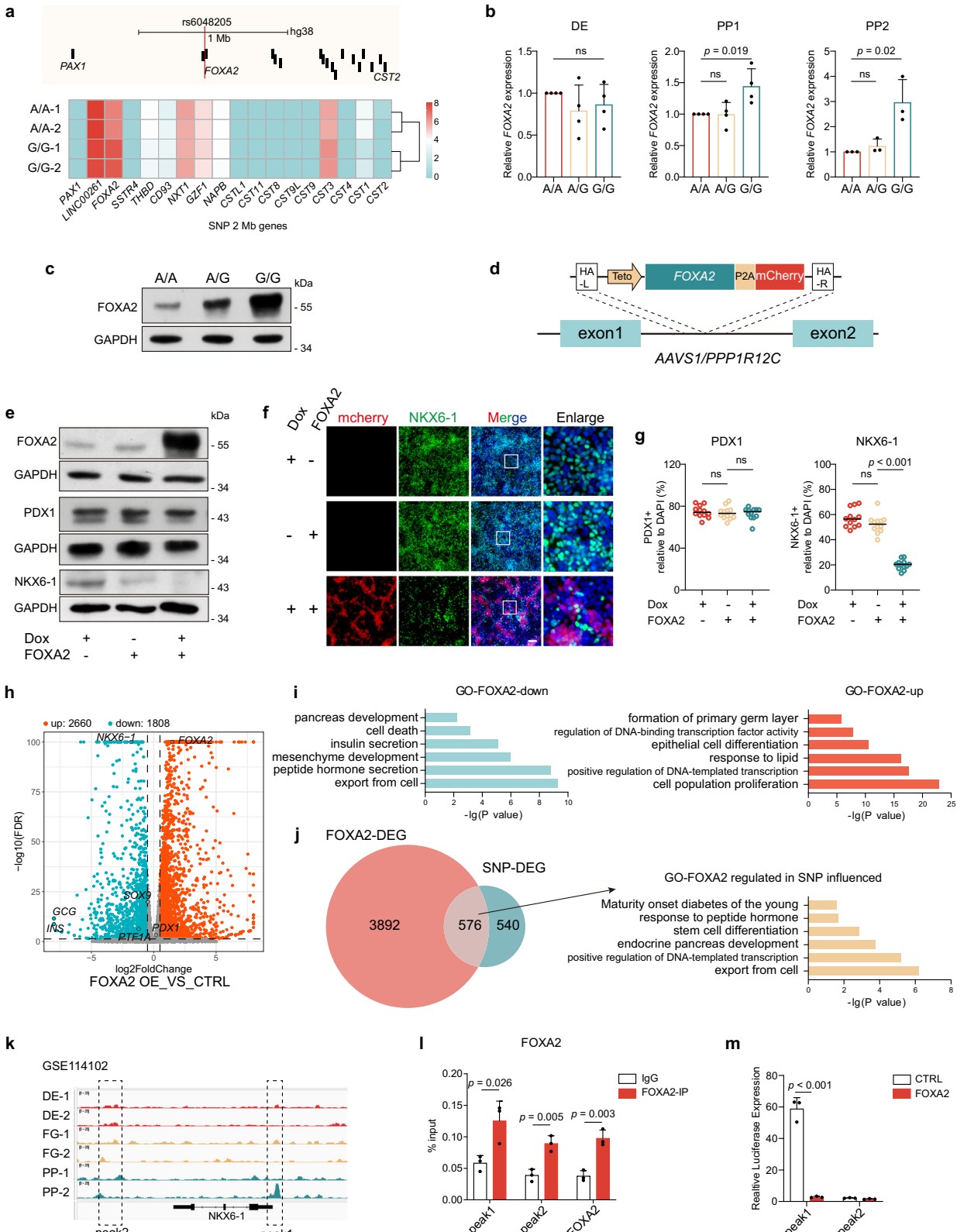

low dose of STZ on the mice with different genotypes (Fig. 5h). After STZ injection, allele G mice exhibited more severe glucose intolerance (Fig. 5i). Importantly, random blood glucose gradually increased for a few days (Fig. 5j). Moreover, the fasting blood glucose in allele G mice appeared significantly increased than the wild-type mice (Supplementary Fig. 7g). The immunostaining data indicated that islet structure was damaged in allele G mutants. Specifically, β

cell number was dramatically decreased along with increased α cells (Fig. 5k, l).

Meanwhile, HFD feeding experiments (Fig. 5h) revealed comparable random blood glucose (Supplementary Fig. 7h), but the fasting blood glucose in allele G mutants was significantly increased after 25 weeks' feeding compared to wild-type controls (Fig. 5m). Similarly, allele G mice exhibited higher glucose intolerance (Fig. 5n), together

**Fig. 3 | rs6048205-G affected pancreatic differentiation through FOXA2 expression. a** Upper: Schematic diagram showing the genes located within 2 Mb away from the rs6048205 locus. Lower: Heatmap showing the expression of these rs6048205-adjacent genes detected by RNA-seq. **b** mRNA levels of *FOXA2* in DE, PP1, PP2 stage detected by RT-qPCR. Gene expression levels were normalized to *GAPDH* and then further normalized to the A/A group (*n* = 4 independent experiments in DE and PP1, *n* = 3 independent experiments in PP2). **c** Protein levels of FOXA2 in PP2 stage detected by Western blot (*n* = 3 independent experiments). GAPDH was used as a loading control. **d** Schematic diagram showing the construction of dox-inducible FOXA2 overexpression system in AAVS1 locus. **e** Protein levels of FOXA2, PDX1 and NKX6-1 in PP2 stage detected by Western blot (*n* = 3 independent experiments). GAPDH was used as a loading control. **f** Representative immunofluorescence images of PP2 cells stained by NKX6-1 (Green). Scale bar, 100 μm. **g** Quantitative analysis for (**f**) and (Supplementary Fig. 5e) (*n* = 12 images). **h** Scatterplot showing differentially expressed genes identified by RNA-seq of PP2 cells of FOXA2 with and without dox (*n* = 2 independent samples). Up-regulated and down-regulated genes upon FOXA2 overexpression were shown in red and blue, respectively. **i** GO analysis of the up- and down-regulated genes in FOXA2 overexpressed PP2 cells. **j** Left: Venn diagram showing the overlap of differentially expressed genes (DEGs) in SNP-G affected genes and DEGs upon FOXA2 over-express; Right: GO analysis of these overlapped genes. **k** The visualization of FOXA2 binding at *NKX6-1* gene locus. A dotted box shows a potentially binding site analyzed from ChIP-seq data (GSE114102). **l** ChIP-qPCR analysis for FOXA2-binding in PP2 cells at *NKX6-1* locus (*n* = 3 independent experiments). **m** Firefly luciferase/ Renilla luciferase results showing the regulation of FOXA2 on *NKX6-1* regulatory elements related to (**k**) (*n* = 3 independent experiments). Data are presented as the mean ± SD. Statistical significance was determined using the unpaired, two-tailed *t*-test in (**a, b, g, l, m**) (ns not significant), one-sided hypergeometric test with multiple comparison adjustments was used in (**i, j**). Source data are provided as a Source Data file.

with dramatically decreased β cell number evidenced by immunostaining data (Supplementary Fig. 7i, j). These results further supported the causal role of rs6048205-G in fasting-glucose trait and diabetes susceptibility.

## Discussion

In this study, we have demonstrated that the noncoding rs6048205 risk allele G decreased the differentiation efficiency of PDX1+ /NKX6-1+ pancreatic progenitor cells (Fig. 1) and led to an abnormal increase of polyhormonal endocrinal cells (Fig. 2), resulting the functional impairment and diabetes susceptibility, through genome-edited human PSC and mouse models. The rs6048205 is in a regulatory region of *FOXA2* locus, and the risk allele G enhanced RXRA binding and activated *FOXA2* expression specifically in pancreatic progenitor stage (Fig. 4). Consistent with this notion, conditionally overexpressed FOXA2 in pancreatic progenitor stage phenocopied the allele G including the reduced NKX6-1-positive cells and defected β cell differentiation (Fig. 3). More importantly, mice carrying the allele G exhibited damaged islets structure and thus were susceptible to HFD or STZ to be hyperglycemia (Fig. 5). Our study illustrated a causal variant for pancreatic differentiation and diabetes susceptibility (Fig. 6), providing new insights for the pathogenesis of diabetes.

FOXA2 is a pioneer factor that could bind to nucleosomal DNA and activate enhancers to regulate endodermal lineage development[38,45]. During pancreatic differentiation, FOXA2 was widely expressed since endoderm formation. Patients with *FOXA2* heterozygous loss-of-function mutation suffered from congenital hyperinsulinism along with endoderm-derived organ malformations[46]. Human PSCs with ablated FOXA2 failed to form pancreatic progenitors and could not eventually generate functional pancreatic β cells[38,47,48]. In addition, conditional inactivation of *Foxa2* in the endodermal gut tube stage (with *Foxa3*-Cre) resulted in defective α cell terminal differentiation[49]. In contrast, ablation of *Foxa2* in fetal β cells (with *Ins*-Cre) disrupted islets morphology and secretory function. It resulted in hyperinsulinemic hypoglycemia[50], indicating that FOXA2 plays stage-specific roles during pancreatic development. Our study showed that stage-specific overexpression of FOXA2 during the pancreatic progenitor stage did not affect the expression levels of the early pancreatic progenitor marker gene PDX1 but resulted in the downregulation of the late pancreatic progenitor marker gene NKX6-1.

As a pioneer factor, FOXA2 primarily binds to super-enhancers in the pancreatic progenitor stage. It is critical for initiating the expression of endocrine lineage genes, suggesting that FOXA2 is indispensable for pancreatic progenitor development[48]. Our finding that FOXA2 overexpression suppresses NKX6-1 indicated that FOXA2 expression level at different stages was critical for pancreatic development. Additionally, we found that the rs6048205-G variant over-activated *FOXA2* expression and resulted in more Glucagon-positive α cells (Supplementary Fig. 4a), consistent with a previous work[49].

FOXA2 can directly bind to the G2 and G1 elements on the *Gcg* gene to activate *Gcg* transcription[51], and in mice with endoderm-specific *Foxa2* knockout (with *Foxa3*-Cre), the number of mature α cells is drastically reduced[49]. The loss of β-cell-specific transcription factors PDX1, PAX4, and NKX6-1 can also elevate *Gcg* gene expression[52–54]. NKX6-1 directly interacts with the *Gcg* promoter and competes with PAX6 to inhibit *Gcg* expression[52]. This suggests that the increased proportion of α cells observed in our study, caused by rs6048205-G, may be attributed to FOXA2 upregulation, which in turn activates *GCG* expression, as well as the downregulation of NKX6-1 or other pancreatic progenitor-related transcription factors.

In addition to the increase in GCG, our study also observed increased SST expression levels in G variant cells (Supplementary Fig. 4a). Previous studies have indicated that the *Nkx6-1* decline in rats contributes to β-cell *trans*-differentiation into δ-cells[55], indicating that the risk variant G can block the development of β cells, leading to a preference for other endocrinal lineages. Interestingly, despite defects in PDX1+ /NKX6-1+ pancreatic progenitors, the G variant increased INS-positive cells, though most of these cells were polyhormonal. Although NKX6-1 expression was reduced during the PP stage in G variant cells, it eventually recovered in the late β-like stage, suggesting a delayed expression pattern. Delayed expression of NKX6-1 was reported to result in the formation of polyhormonal cells[56], which is consistent with our findings that the G mutation leads to an increase in polyhormonal cells. Although NKX6-1 can directly bind to and regulate β-cell-specific transcription factors and insulin processing genes in β-cells[57], the restricted temporal window for its expression limits its ability to fully correct these developmental defects to prevent the polyhormonal cell fate. Besides, previous studies have reported that while the loss of *Nkx6-1* significantly reduces the number of β cells in mouse islets, a small number of β cells are still produced[58,59], indicating that other factors may also contribute to the generation of INS-expressing cells. Upon revisiting our SNP PP2 RNA-seq data, we observed that while *NKX6-1*, *PTF1A*, and *SOX9* were downregulated in the G variant pancreatic progenitors, critical pancreatic transcription factors such as *NKX6-2* and *MNX1* were up-regulated. NKX6-2 was reported to compensate NKX6-1 in regulating β-cell fate[59,60], and MNX1 was essential for pancreatic β-cell development and identity maintenance[61,62]. Thus, despite reduced NKX6-1 expression in pancreatic progenitors with the G variant, the upregulation of these transcription factors may play a compensate role to activate insulin expression.

The RXR superfamily regulates various biological processes, including differentiation, cellular growth, and metabolism[63]. RXRs play a unique and central role within this superfamily by serving as obligatory heterodimeric partners for several key receptors, such as retinoic acid receptors (RARs), thyroid hormone receptors, vitamin D receptors, peroxisome proliferator-activated receptors (PPARs), liver X receptors (LXRs), and others, thereby influencing their activity[64],

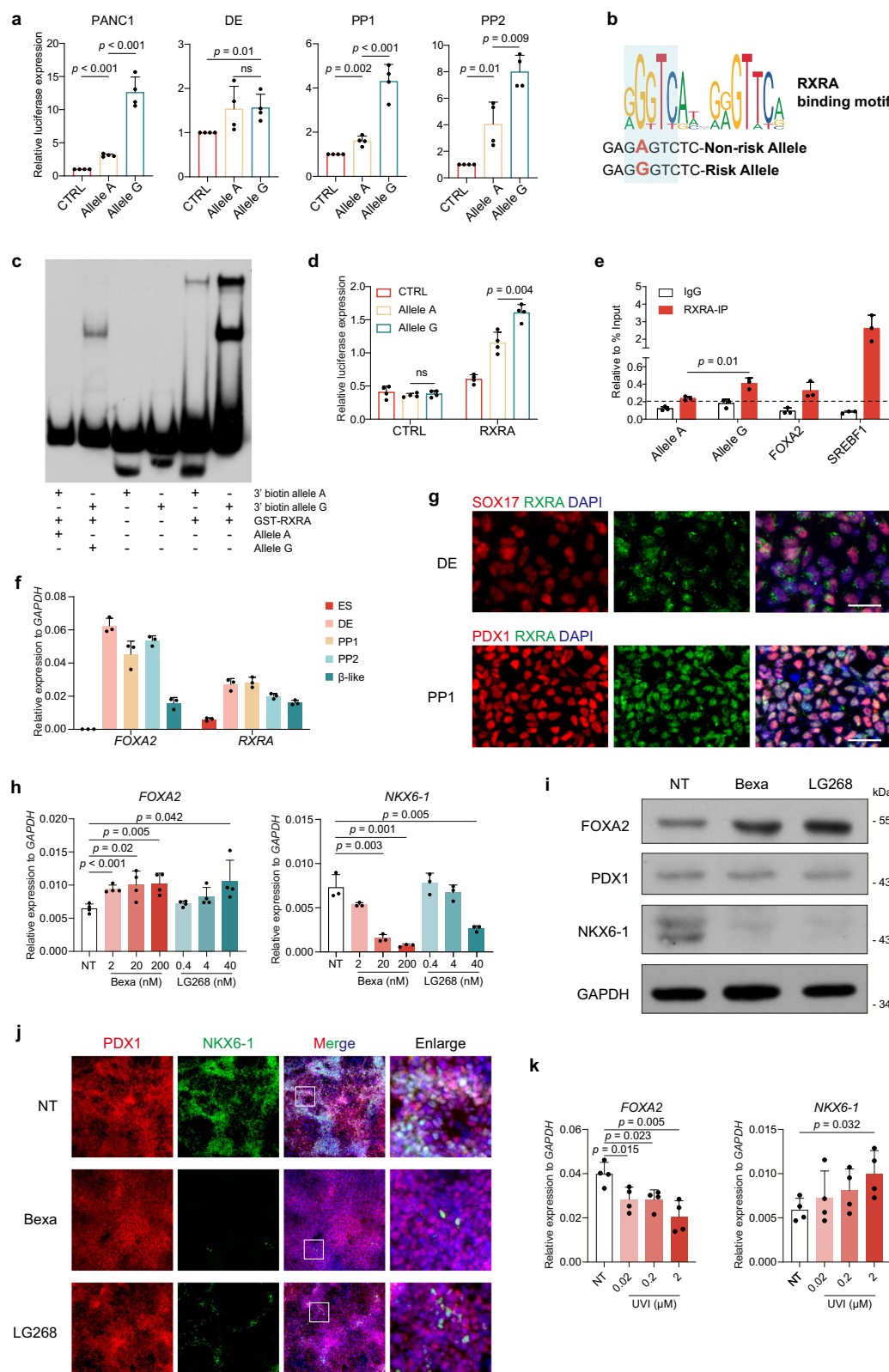

which partner cooperated with RXRA to regulate *FOXA2* is waiting for further exploration. Xu and colleagues reported that RA inhibited the differentiation of FOXA2-positive floor plate cells and promoted neural differentiation[65]. RA plays a crucial role in the induction of PDX1 expression and pancreatic progenitor differentiation[19,66,67], and TTNPB was used in our differentiation protocol to activate RA. However, whether RA forms dimers with RXRA to regulate FOXA2 in an SNP rs6048205 G variant-dependent manner requires further investigation.

In addition, we recognize the limitations associated with our RXR compound-based studies (Fig. 4, Supplementary Fig. 6) as none of these compounds can distinct the three subtypes of the RXR superfamily, which includes RXRA, RXRB, and RXRG. The compounds used in this study, including bexarotene, LG00268, and UVI3003, target all

**Fig. 4 | rs6048205-G enhanced RXRA binding to activate FOXA2 expression in pancreatic progenitor stage abnormally. a** Firefly luciferase/Renilla luciferase results of rs6048205 alleles in PANC1, DE, PP1, PP2 cells (*n* = 4 independent experiments). **b** The sequence around rs6048205 and the consensus RXRA binding motif from JASPAR. **c** EMSA results show purified RXRA protein binding to biotin-labeled probes containing two allelic variants of rs6048205 (*n* = 3 independent experiments). **d** Firefly luciferase/Renilla luciferase results show RXRA over-expression regulation on rs6048205 in 293T (*n* = 4 independent experiments). **e** Allele-specific ChIP-qPCR assay for comparing RXRA binding between allele A and allele G in Huh7 cells, which is heterozygous in rs6048205 (*n* = 3 independent experiments). **f** Time-course quantification of relative mRNA levels of *RXRA* along with *FOXA2* detected by RT-qPCR analysis (*n* = 3 independent experiments). **g** Representative immunofluorescence images of DE and PP cells stained by RXRA with SOX17 and PDX1, respectively (*n* = 3 independent experiments). Scale bar,

50 μm. **h** mRNA levels of *FOXA2* (*n* = 4 independent experiments) and *NKX6-1* (*n* = 3 independent experiments) in PP2 cells treated with different dosages of RXR agonists bexarotene (Bexa) and LG000268 (LG268) detected by RT-qPCR analysis. **i** Protein levels of FOXA2, PDX1, and NKX6-1 in PP2 cells treated with the highest dosage of RXR agonists bexarotene (Bexa, 200 nM) and LG000268 (LG268, 40 nM) detected by Western blot (*n* = 3 independent experiments). GAPDH was used as a loading control. **j** Representative immunofluorescence images of PDX1 and NKX6-1 staining in PP2 cells treated with the highest dosage of RXR agonists bexarotene (Bexa, 200 nM) and LG000268 (LG268, 40 nM). **k** mRNA levels of *FOXA2* and *NKX6-1* in G/G PP2 cells treated with different dosage of RXR inhibitor UVI3003 (UVI) detected by RT-qPCR analysis (*n* = 4 independent experiments). Data are presented as the mean ± SD. Statistical significance was determined using the unpaired, two-tailed *t*-test in (**a, d, e, h, k**) (ns not significant). Source data are provided as a Source Data file.

three RXR subtypes with comparable EC50 values. Due to the high structural similarity of RXRs, developing subtype-specific activators or inhibitors remains a challenge[68,69]. Nevertheless, our results support that RXRA activation can regulate FOXA2 expression through the SNP, although we cannot exclude the potential roles of RXRB and RXRG from this process.

In summary, we have identified a gain-of-function noncoding variant that causes defective pancreatic development and contributes to diabetes susceptibility. Meanwhile, we have dissected the pathogenic mechanism through a RXRA-*FOXA2*-*NKX6-1* manner. In addition, our study provides a paradigm to explore the mechanisms of diabetes caused by noncoding mutation.

# Methods
## Ethics
All experimental procedures were approved by the Biomedical Ethics Committee of Wuhan University (WHU-LFMD-IRB2024026) and the Animal Care and Ethical Committee of Medical Research Institute, Wuhan University (MLIC202007).

## Cell culture and differentiation
Human ESC line HUES8 and iPSC line PGP1 were used in this study. These PSCs were cultured in mTeSR1 medium (STEMCELL Technologies, Cat#85850) on Matrigel-coated plates at 37 °C with 5% $CO_2$. Human embryonic kidney 293T (HEK293T) cells, human pancreatic cancer cells (PANC1), and human liver cell line Huh7 were cultured in DMEM medium supplemented with 10% fetal bovine serum (FBS) (Gibco, Cat#10100147) and 1% penicillin-streptomycin (PS) (Gibco, Cat#10378016) at 37° C with 5% $CO_2$.

Genome-edited cell lines were analyzed in parallel with controls in all differentiation assays. For pancreatic β cell differentiation, a published differentiation protocol[33] was used here with certain modifications. Briefly, human PSCs were isolated into single cells using Accutase (STEMCELL Technologies, Cat#07922) and seeded at growth-factor-reduced Matrigel-coated plates with 30,000 to 50,000 cells/cm$^2$ density. For induction of definitive endoderm, plated cells were cultured in 1:1 IMDM (Gibco, Cat#C12440500BT) and F12 (Gibco, Cat#C11765500BT) medium (IF12) supplemented with 1% BSA (YEASEN, Cat#36101ES76), 100 ng/mL Activin A (PeproTech, Cat#120-14 P) and 2.5 μM CHIR99021 (Selleck, Cat#S2924) for 1 day. The CHIR99021 was removed from the medium for the following 2–3 days. For induction of PDX1+ pancreatic progenitors (PP1), obtained DE cells were cultured in MCDB131 (Sigma, Cat#10372019) supplemented with 1.5 g/L sodium bicarbonate (Gibco, Cat#25080094), 10 mM glucose (Sigma, Cat#G7528), 2% BSA, 1 × ITS-X (BasalMedia, Cat#S452J7), 0.25 mM ascorbic acid (Gibco, Cat#A5960), 50 ng/mL KGF (Pepro-Tech, Cat#100-19-500), 0.25 μM SANT1 (Selleck, Cat#S7092), 100 nM TTNPB (Selleck, Cat#S4627), 500 nM PDBU (Sigma, Cat#P1269) and 200 nM LDN193189 (Selleck, Cat#S7507). For induction of PDX1+/ NKX6-1+ pancreatic progenitors (PP2), obtained PP1 cells were

cultured in MCDB131 supplemented with 1.5 g/L sodium bicarbonate, 10 mM glucose, 2% BSA, 1 × ITS-X, 0.25 mM ascorbic acid, 2 ng/mL KGF, 0.25 μM SANT1, 10 nM TTNPB, 250 nM PDBU, 400 nM LDN193189 and 2 μM IWR1 (Selleck, Cat#S7086). For induction of pancreatic endocrine progenitors (PEP), PP2 cells were cultured in MCDB131 supplemented with 1.5 g/L sodium bicarbonate, 20 mM glucose, 2% BSA, 1 × ITS-X, 0.25 mM ascorbic acid, 10 μM Repsox (Selleck, Cat#S7223), 10 μg/mL heparin (Selleck, Cat#S1346), 0.25 μM SANT1, 5 nM TTNPB, 200 nM LDN193189, 1 μM T3 (Sigma, Cat#64245) and 10 μM Zinc sulfate (Sigma, Cat#Z0251). For induction of mature β cells, obtained PEP cells were cultured in MCDB131 with 20 mM glucose and 2% BSA, 1.5 g/L sodium bicarbonate, 0.05 mM ascorbic acid, 1 × ITS-X, 1 × GlutaMAX, and 1% PS. In addition, 200 nM LDN193189, 1 μM T3, 10 μM Repsox, 10 μM YO-01027 (Selleck, Cat#S2711), and 10 μM zinc sulfate were added for the first 3 days, 200 nM LDN193189, 1 μM T3, 10 μM Repsox, 10 μg/mL heparin, and 10 μM zinc sulfate were added for the next 3–7 days, 1 μM T3, 10 μM Repsox, 10 μM Trolox (EMD, Cat#648471), 10 μg/mL heparin sodium, 2 μM R428 (APExBIO, Cat#A8329), 10 μM zinc sulfate, and 10 mM N-cys (Sigma, Cat#A9165) for the last 3–7 days.

## CRISPR/Cas9-mediated knock-in in human PSCs
The single guide RNA (sgRNA) (CCCCAACTCTACTCATTGTG) was constructed in pX459 expression plasmid. 507 bp double-stranded donor DNA was gel-purified from PCR products of genome sequence carrying allele-G. One million PGP1 or HUES8 cells were electroporated with 5 μg donor DNA, 4 μg pX459 plasmid expressing guide RNA, and 0.5 μg pMAX-GFP for control using LONZA P3 kit. Then, electroporated cells were seeded on Matrigel-coated plates in mTeSR1 medium supplement with 10 μM Y-27632 (Selleck, Cat#S1049) and 100 ng/ mL nocodazole (Selleck, Cat#S2775)[70]. After 36 hours of culture, GFP-positive single cells were sorted using BD FACS Aria III and replated on Matrigel-coated 96-well plates with one cell per well. The generated colonies were expanded and genotyped by Sanger sequencing.

## Generation of inducible FOXA2-overexpressing cell line
*FOXA2* CDS sequence was obtained from DE cDNA. The dCas9 sequence of Gen1 vector (pAAVS1-NDi-CRISPRi, Addgene, Cat#73497) was replaced with *FOXA2* CDS. Gen1-FOXA2 and pX459 (pSpCas9(BB)−2A-Puro, Addgene, Cat#48139) containing sgRNA targeting *AAVS1* locus were electroporated into HUES8 cells. After 24 hours' culture, positive cells were purified by 100 ng/mL G418 (SANTA CRUZ, Cat#sc-29065A) for about 14 days.

## Mice manipulation
Knock-in C57BL/6 J mice carrying the allele-G were constructed commercially (Shanghai Model Organisms Center Inc.) and genotyped by Sanger sequencing. Mice were maintained at room temperature (21 ± 2 °C), with a humidity of 50% ± 15% and a 12 hours' cycle of light and dark in the animal facility of Medical Research Institute, Wuhan

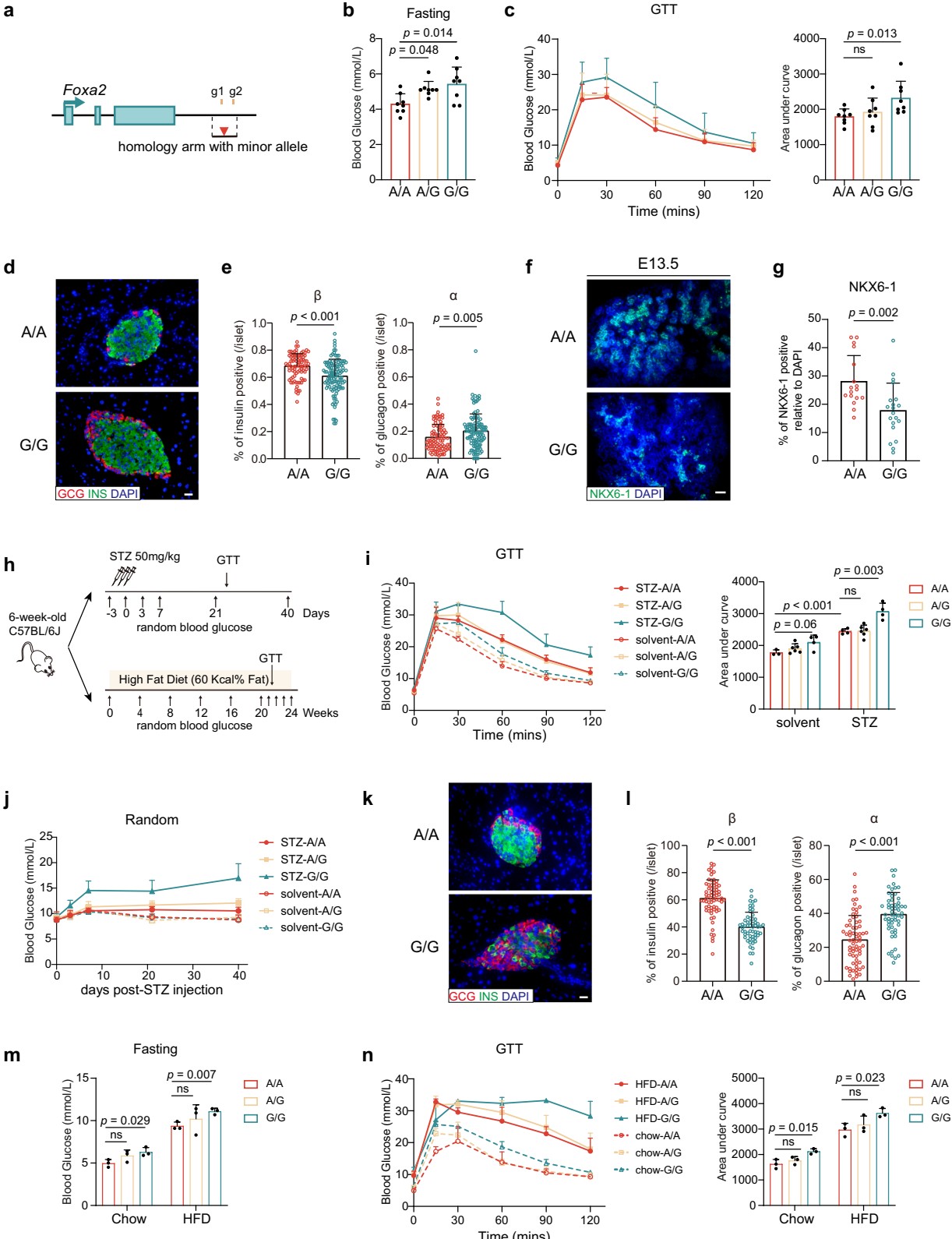

University. A regular chow diet (BEIJING KEAO XIELI FEED CO., LTD., Cat#1016706476803973120) or a high-fat diet (fat content, 60% kcal%, Research Diets, Cat#D12492i) was applied with free access to water. Only male mice were used to eliminate the interference of estrous cycle and hormonal fluctuations in female mice and no sex differences in experimental outcomes were noted.

**Immunofluorescence staining**

Cells were washed with PBS and fixed in 4% paraformaldehyde at room temperature and then were blocked and permeabilized by block buffer (PBS with 10% (v/v) donkey serum and 0.3% Triton X-100) for 1 hour after PBS washing. Primary antibodies and secondary fluorescent antibodies were diluted in block buffer at proper concentration. Cells

**Fig. 5 | rs6048205-G regulated blood glucose homeostasis in vivo. a** Schematic diagram of generation of rs6048205 knock-in mice by CRISPR/Cas9. The guide RNA targeting site was shown in yellow, and the SNP location was in red. **b** Fasting blood glucose levels in mice with different genotypes (*n* = 8 mice). **c** Glucose tolerance tests (GTT) for mice with different genotypes. Area under the curve (AUC) results was shown on the right (*n* = 8 mice). **d, e** Representative immunofluorescence staining of adult mice islets with insulin (INS) and glucagon (GCG) (**d**) and statistical analysis (**e**) of β cells and α cells percentage in each islet (*n* = 89 in A/A, *n* = 115 in G/G). Scale bar, 20 μm. **f, g** Representative immunofluorescence staining of E13.5 pancreas with NKX6-1 (**f**) and statistical analysis (**g**) of NKX6-1 positive percentage in each field under the microscope (*n* = 17 in A/A, *n* = 20 in G/G). Scale bar, 50 μm. **h** Schematic of experimental design of STZ (Streptozotocin) injection and high-fat diet feeding. **i** GTT result of mice treated with STZ or citrate

solvent. AUC was shown on the right (*n* = 4 mice in group STZ A/A, STZ G/G and solvent G/G, *n* = 6 in STZ A/G and solvent A/G, *n* = 3 in solvent A/A, *n* = 6 in solvent A/G). **j** Quantification of the random blood glucose after STZ injection (*n* = 4 mice in STZ A/A, STZ G/G and solvent G/G, *n* = 6 in STZ A/G and solvent A/G, *n* = 3 in solvent A/A, *n* = 6 in solvent A/G). **k, l** Representative immunofluorescence staining of mice islets treated with STZ (**k**) and statistical analysis of β cells and α cells percentage in each islet (**l**) (*n* = 68 in A/A, *n* = 58 in G/G). Scale bar, 20 μm. **m** Fasting blood glucose levels after HFD feeding (*n* = 3 mice). **n** GTT result of HFD-fed mice of each genotype. AUC was shown on the right (*n* = 3 mice). Data are presented as the mean ± SD. Statistical significance was determined using the unpaired, two-tailed *t*-test in (**b, c, e, g, i, j, l–n**) (ns not significant). Source data are provided as a Source Data file.

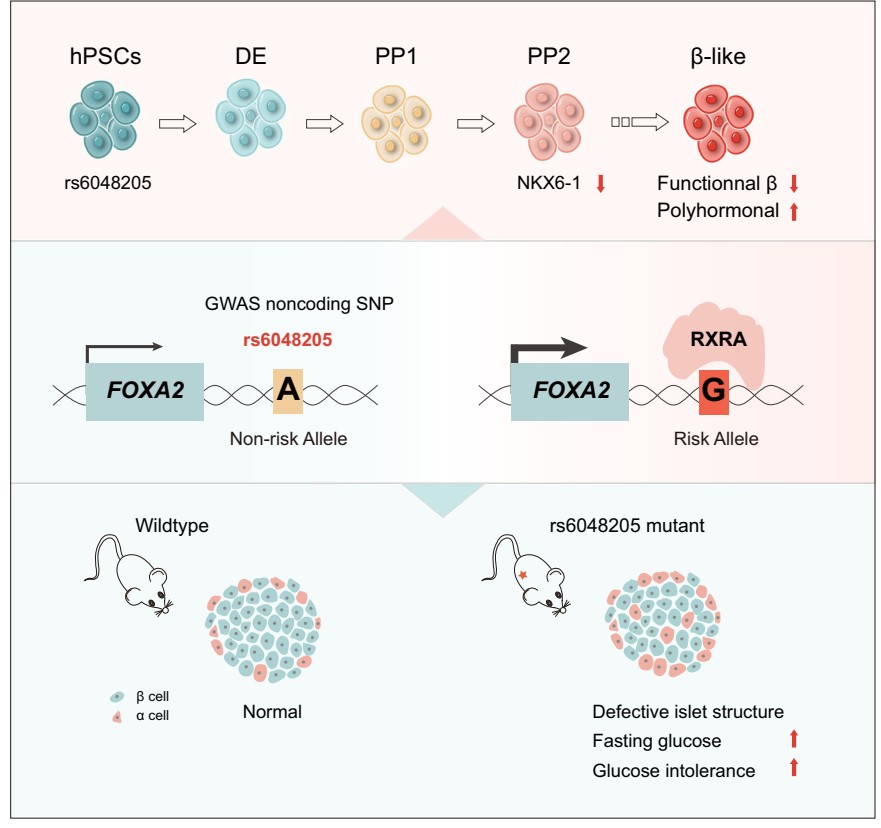

**Fig. 6 | Model for the role of rs6048205 in pancreatic differentiation and glycemia homeostasis.** The noncoding SNP rs6048205 risk allele G enhances RXRA binding and activates FOXA2 expression, specifically in the pancreatic progenitor stage. The rs6048205 risk allele G reduces the differentiation efficiency of NKX6-1+ pancreatic progenitor cells, leading to a decrease in functional β cells and an abnormal increase in polyhormonal endocrine cells in vitro. Mice carrying the risk allele G exhibit higher fasting glucose levels and impaired glucose tolerance, contributing to increased susceptibility to diabetes in vivo.

were incubated with diluted primary antibody overnight at 4 °C. After washing with PBS three times, and cells were incubated with diluted corresponding secondary fluorescent antibodies and incubated at room temperature in the dark for 2 h. The nuclei were counterstained with 2‰ 4,6-diamidino-2-phenylindole (DAPI, Sigma, Cat#10236276001). The results of immunofluorescence were visualized and imaged under an Olympus IX53 microscope. The antibodies for immunofluorescence assay used in present work include: anti-SOX17 (1:200, R&D, Cat#AF1924), anti-OCT4 (1:200, SANTA CRUZ, Cat#sc-5279), anti-PDX1 (1:200, R&D, Cat#AF2419), anti-NKX6-1 (1:300, ABclonal, Cat#A20419), anti-NKX6-1 (1:400, Cell Signaling Technology, Cat#54551), anti-FOXA2 (1:200, R&D, Cat#AF2400), anti-RXRA (1:300, Cell Signaling Technology, Cat#3085), anti-C-peptide (1:400, DSHB, Cat#GN-ID4), anti-Glucagon (1:500, Servicebio, Cat#GB11097), anti-

Glucagon (1:500, Servicebio, Cat#GB11335), anti-SST (1:400, ZAGC-BIO, Cat#EP130/ZA-0567).

### Flow cytometry

Cells were digested into single cells by 0.05% trypsin-EDTA and suspended in PBS containing 2% FBS. Cells were collected by centrifuging at 300 g for 5 minutes at 4 °C, and then re-suspended and fixed according to the manufacturer's instructions of Transcription Factor Buffer Set (BD, Cat#562574). Fixed cells were washed and incubated with diluted primary antibodies overnight at 4 °C. Then the cells were washed and incubated with diluted secondary fluorescent antibodies in the dark for 2 hours at room temperature. Primary antibodies and secondary fluorescent antibodies were diluted in 1x Perm/Wash solution. After wash, cells were resuspended in PBS and analyzed with a

flow cytometer (BD LSRFortessaX20 or ACEA NovoCyte). Representative flow cytometry pseudocolor plots and gating strategy are shown in Supplementary Fig. 8. The antibodies for immunofluorescence assay used in present work include: anti-SOX17 (1:500, R&D, Cat#AF1924), anti-PDX1 (1:500, R&D, Cat#AF2419), anti-NKX6-1 (1:500, DSHB, Cat# F64A6B4), anti-C-peptide (1:800, DSHB, Cat#GN-ID4), anti-Glucagon (1:800, Servicebio, Cat#GB11097).

## RT-qPCR
Total RNA was extracted using Hipure Total RNA Mini Kit (Magen) according to the manufacturer's instructions. 1 µg RNA was reverse-transcribed into cDNA using the ABScript III RT Master Mix (ABclonal, Cat#RK20428). qPCR reactions were performed using 2× Universal SYBR Green Fast qPCR Mix (ABclonal, Cat #RK21203) on the CFX384 Touch Real-Time PCR Detection System (Bio-Rad). The expression levels of different mRNAs were normalized to *GAPDH*. The primers used in the RT-qPCR assays are listed in Supplementary Table 1.

## Dual luciferase reporter assay
Sequences containing different allele of rs6048205 (A or G) were cloned into pGL-4.17 [*luc2*/Neo] vector (Promega, Cat#9PIE672). Cells were digested and replated in 24-well plates before transfection. After the cells reached 70%–80% confluence, constructed pGL-4.17 plasmid was respectively transfected along with pRL-TK through LIPO-plus (Sage, Cat#Q03004). Cells were lysed and measured after 24 hours according to the manufacturer's instructions for the Dual-Luciferase Reporter Assay System (Promega, Cat#E1910). For luciferase assay with gene overexpression, CDS was constructed into pCMV3Tag8li and transfected with PGL4.17 and pRL-TK into 293T cells, while empty pCMV3Tag8li vector was used as negative control.

## Western blot
Cell pellets were lysed using RIPA buffer (Beyotime, Cat#P0013C) with a cocktail (Roche, Cat#4693132001) for 30 minutes at 4 °C. The lysates were centrifuged, and the supernatant was collected. Proteins were separated by 10% SDS-PAGE and transferred to the nitrocellulose membrane. The membranes were washed by TBST and blocked by 5% (w/v) skimmed milk in TBST for 30 minutes at room temperature. After blocking, the membranes were incubated with the indicated antibodies overnight at 4 °C or 2 hours at room temperature. Then, the membranes were washed three times with TBST and incubated with HRP-conjugated second antibodies for 1 hour at room temperature. After washing three times with TBST, the membranes were incubated with ECL (Millipore, Cat#WBUSLS0100) in the dark and visualized using a film image system in the dark room. Antibodies were diluted using 5% (w/v) skimmed milk in TBST. The antibodies for western blot used in the present work include: anti-NKX6-1 (1:500, DSHB, Cat#F64A6B4), anti-PDX1 (1:500, R&D, Cat#AF2419), anti-FOXA2 (1:1500, Cell Signaling Technology, Cat#8186), anti-GAPDH (1:20000, ABClonal, Cat#AC002).

## ChIP-qPCR
Cells for ChIP were cultured on 10-cm dish. These cells were washed with PBS and crosslinked with 1% formaldehyde for 10 minutes at room temperature, then 0.125 M glycine for 5 minutes was used to stop the crosslink. After being washed with cold PBS, cells were collected by cell scraper and centrifuged at 4 °C, then the cells were lysed with Cell Lysis Buffer (10 mM Tris-HCl pH 8.0, 140 mM NaCl, 0.2% NP-40) with a cocktail. After centrifuge, the supernatant was discarded, and the nuclei were collected and resuspended in nuclear lysis buffer (50 mM Tris-HCl (pH 8.0), 10 mM EDTA (pH 8.0), 1% SDS) and sonicated by Diagenode Bioruptor Pico for ten cycles with 30 s/on and 30 s/off. Then, the supernatants were collected and incubated with antibody overnight at 4 °C. Protein A/G Magnetic Beads were added the next day and incubated for 2 hours at room temperature, and then the DNA fragments were eluted and reverse-crosslinked at

67 °C for 5 hours. After that, immunoprecipitated DNA were purified by HiPure Gel DNA Mini Kit (Magen, Cat#D2111), and the qPCR was performed. The primers used in the ChIP-qPCR assays are listed in Supplementary Table 2.

## Electrophoretic mobility shift assay (EMSA)
*RXRA* CDS was inserted into pGEX4T-1 vector containing GST tag, and protein expression and purification were performed as before[71]. EMSA was performed according to the manufacturer's instructions for Chemiluminescent EMSA Kit (Beyotime, Cat#GS009). Briefly, oligonucleotide probes with different alleles of rs6048205 were synthesized and labeled with the biotin by Sangon Biotech (Wuhan, China). Purified RXRA protein was incubated with biotin-labeled oligos in a binding buffer, while unlabeled oligos and reactions without RXRA were used as negative controls. The reaction mixtures were resolved on 4% non-denaturing polyacrylamide gels and transferred to a nylon membrane. Then, the DNA oligomers were crosslinked to a membrane by UV cross-linker, detected by BeyoECL, and visualized using a film image system in the dark room. Probes used for EMSA assays are listed in Supplementary Table 3.

## Glucose tolerance test (GTT) and insulin tolerance test (ITT)
Before GTT, mice were fasted for 14 hours. Fasting glucose was measured by glucometer (Yuwell, Yuezhun type 2). Then, mice were injected intraperitoneally with glucose (2 g/kg body weight), and the blood glucose were measured at 15, 30, 60, 90, 120 minutes after glucose injection.

Before ITT, mice were fasted for 6 hours, then mice were injected intraperitoneally with human insulin (0.75 U/kg body weight; Humalog, Eli Lilly), and the blood glucose were measured before insulin injection and 15, 30, 60, 90, 120 minutes after injection.

## Streptozotocin (STZ) treatment
STZ was freshly dissolved in citrate buffer (10 mM, pH 4.2) before injection. 6–8-week-old male littermates were fasted overnight and injected intraperitoneally with STZ (50 mg/kg body weight, Sigma, Cat#V900890) or citrate buffer once per day for three consecutive days. Blood glucose was measured 72 hours after the last injection.

## High-fat diet (HFD) feeding
Feedings of male littermates were switched to a high-fat diet (HFD; fat content, 60% kcal) (Research Diets, Cat#D12492i) from 6 to 8 weeks old. Blood glucose and body weight were measured weekly.

## Cell apoptosis analyses
The Annexin V-FITC/PI cell apoptosis detection kit (YEASEN, Cat#40302ES60) was used for apoptosis analysis. About $1 \times 10^5$ single cells were resuspended by binding buffer supplemented with Annexin V-FITC and PI at room temperature for 10 min. Finally, a FACS Cytoflex flow cytometer was used to analyze these cells.

## RNA-seq and data analysis
Total RNA was extracted using HiPure Total Mini Kit and quantified by a DNA/Protein Analyzer (QuaWell, Sunnyvale, California, USA). RNA was sent to YINGZI GENE (Wuhan, China) for RNA-seq library preparation and sequencing. The human genome (hg38) with HISAT2 was used to align RNA seq data. The GENCODE V29 gene transfer format (GTF) was used to count reads by the FeatureCounts (v2.0.1), and we quantified gene expression level with TPM (transcripts per million). Differential gene expression analysis was performed for binary comparisons using the R package DESeq2 with the cutoff of fold-change >1.5 and $p < 0.05$. Gene Ontology analysis was performed using g:Profiler[72] (https://biit.cs.ut.ee/gprofiler/gost). Venn diagram was performed by bioinformatics (https://www.bioinformatics.com.cn), an online data analysis and visualization platform.

## Statistics and reproducibility

All experiments were performed independently at least three times, except for RNA-seq, which was done in two independent replicates. The number of mice used for each group was usually more than three. Data was presented as the mean ± SD. All statistical analysis was performed by GraphPad Prism 9 software, and the significance level was calculated by student's unpaired $t$-test (Two-tailed). Statistical significance is indicated in each figure.

## Reporting summary

Further information on research design is available in the Nature Portfolio Reporting Summary linked to this article.

## Data availability

The RNA-seq data generated in this study have been deposited in the Gene Expression Omnibus (GEO) database under accession code GSE249854. The processed ChIP-seq data are available in GEO database under the accession code GSE114102, GSE131817, GSE148368, GSE104840. Uncropped Western blots and raw data to generate all graphs within the Figures and Supplementary Figs. are provided as a Source Data File. Source data are provided with this paper.

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

## Acknowledgements

We thank the Core Facility and the Animal Facility of Medical Research Institute of Wuhan University for technical support. We thank Dr. Yi Zhang at Boston Children's Hospital and Dr. Donghui Zhang at Hubei University for insightful discussion, Drs. Bishi Fu and Yucheng Xia at Wuhan University for providing plasmid or cells. We thank Pei Lu, Jie Yang, and other laboratory members for their technical help and discussion. This work was supported by the National Natural Science Foundation of China (31970608 and 32350019 to Wei Jiang), the Natural Science Foundation of Hubei Province - Innovation Group project (2024AFA018 to Wei Jiang), Wuhan Intellectual Innovation Fund (2023020201010074 to Wei Jiang), Seed Fund Program for Sino-Foreign Joint Scientific Research Platform of Wuhan University (WHUDKUZZJJ202205 to Wei Jiang), and the Fundamental Research Funds for the Central Universities in China (2042022dx0003 to Wei Jiang).

## Author contributions

W.J. (Wei Jiang) conceived and supervised the project and designed the experiment together with Y.L. (Yinglei Li) and L.J.; Y.L. (Yinglei Li) performed most of the bench experiments with the help from R.Z. particularly in ChIP, EMSA and luciferase experiment; C.Y. analyzed the sequencing data; L.J. performed a part of pancreatic differentiation experiment with the help of W.J. (Wenwen Jin), and X.Z.; R.L. provided support in cell culture and animal experiment. L.C assisted with the isolation of the embryonic pancreas. Y.L. (Yuanyuan Luo) helped with the construction of cell lines. J.T and Z.D participated in the design and discussion of the physiological function of the SNP. Y.L. (Yinglei Li) drafted the manuscript, W.J. (Wei Jiang) and Y.L. (Yinglei Li) finalized the manuscript. All authors contributed to and approved the final manuscript.

## Competing interests

The authors declare no competing interests.
