## [Peer Review file · Nature Communications]

A noncoding variant confers pancreatic differentiation defect and contributes to diabetes susceptibility by recruiting RXRA

Corresponding Author: Dr Wei Jiang

Version 0:

Reviewer comments:

Reviewer #1

(Remarks to the Author)

This manuscript studies pancreatic development and the impact of a pathogenic SNP in this process. The authors use the human pluripotent stem cell model to examine a SNP downstream of the FOXA2 gene, generating lines that are heterozygous or homozygous for the SNP. They find the SNP leads to abnormal pancreatic development with decreased pancreatic progenitors and skewed differentiation to a polyhormonal endocrine phenotype. Overexpression studies of FOXA2 also blocks pancreas development and a mouse model of the SNP has defects in glucose homeostasis. Overall, the studies presented are very interesting but there are some issues that need to be addressed. See below for specific comments.

1) The authors only examine a single clone for the human stem cell genome editing. They should show multiple clones to confirm findings are consistent or even better the editing in a second genetically distinct stem cell line would show the effect is independent of genetic background. This is especially important as the differentiation efficiency shown in figure 1 is very low, with most publications in the field demonstrate at least 30-75% NKX6.1+ cells at the PP2 stage. In addition, edited lines should be tested for genome integrity such as karyotype or CNV analysis.

My worry with CRISPR based homozygous gene editing when analysis is based upon sequencing alone is on target large indels/insertions or rearrangements, see pmid: 35276091 . Digital droplet PCR or other methods such as demonstrating another heterozygous SNP is present within the PCR amplicon where the homozygous mutation is introduced to show the homozygous G/G clones have biallelic amplification would be helpful.

2) In figure 2 the examination of polyhormonal cells only looks at INS and GCG. SST is another common hormone which can present in polyhormonal cells. The analysis should include SST cells as well to see if the impact of the variant is only on GCG expression or polyhormonal cells in general. When examining GCG/SST expression the authors should also look at monohormonal GCG and SST populations (alpha and delta cells) to see if these are impacted as well.

There is a concern that flow cytometry and microscopy show drastic differences in hormone expression levels. For example in 2c the G/G line has close to 40% GCG+ cells by microscopy but in 2f its only 3%. Why such a huge discrepancy? I worry one of the assays is off. Flow cytometry plots with isotype controls should be shown in supplemental data.

3) In Figure 3 the authors do not look at FOXA2 at the protein level in the G/G compared to A/A lines. This would be useful to show in addition to RNA.

4) In figure 4 agonists of RXR are given and show FOXA2 expression is increased with decreased differentiation efficiency. RA is present in these differentiations which would activate RXR/RAR as well. Can lowering dose of RA correct the defect in the G/G line or at least drop FOXA2 levels? While not absolutely necessary, rescue experiments are more convincing than small molecule experiments that inhibit differentiation because off targets effects of small molecules are a concern.

5) Its odd that at the PP2 stage the G/G line has a pretty severe decrease in pancreatic progenitors yet makes more INS+ cells at end stage culture though they are skewed to the polyhormonal fraction? This should at least be discussed in the

manuscript.

Minor issues:

1) Grammar and English language issues throughout the manuscript making it difficult to read. Please correct. An editing service could be helpful here.

Reviewer #2

(Remarks to the Author)

The manuscript "A noncoding variant confers pancreatic differentiation defect and contributes to diabetes susceptibility by recruiting RXRa" by Li et al examines the significance of a non coding SNP (rs6048205-G) located downstream of FOXA2 that is associated with increased blood glucose levels and pancreatic beta cell dysfunction using human pluripotent stem cell (hPSC) and mouse models of islet development and function. Using CRISPR, the authors modeled this SNP in human stem cells as WT(A/A), Het (A/G), and HZ (G/G) alleles, with the G allele being the predicted risk allele. There is some novelty focusing on a single non-coding SNP, although others have examined non coding SNPs in the context of diabetes before (PMID: 32442395, 36778047, 33505025). Here, the authors demonstrate a reduced hPSC derived pancreatic islet differentiation capacity in the presence of the A/G and G/G mutants, with G/G having the most severe impacts and leading to increased formation of polyhormonal cells in the beta-like cell differentiations, specifically working through regulation of NKX6.1 and FOXA2. This analysis includes a couple of valuable RNA-sequencing experiments identifying a broad swath of differentially regulated genes in their G/G mutant. The authors go on to suggest that that this A-G mutation creates an RXRa binding site and complete a series of experiments testing this and trying to determine if RXRa could be responsible for regulating FOXA2 during pancreatic progenitor specification. In their last figure, the authors also created a new mouse model that contains this single point mutation and showed that adult mice have increased fasting blood glucose levels and challenging the mice with either STZ depletion of beta cells or a high fat diet, mice carrying the A/G or G/G alleles were more severely impacted compared to A/A controls. While there was quite a bit of effort dedicated to examining this particular SNP and how it impacts the molecular networks that define islet cell differentiation, there are some key controls, experiments, and interpretations of data that are of concern to me:

1) Broadly, is it known that this SNP falls within an enhancer? Are there active enhancers marks that have been shown (H3K27ac or H3K4me1?) Or is this a known open region of chromatin (ATAC-seq)? These datasets are available and should be examined/mentioned here.

a. Similarly, are there any transcripts found in this region (like non-coding RNAs, for example) that could have regulatory implications? How do we know for sure this is a non-coding region?

2) In Figure 1C, the expression of a handful of important beta cell TFs are measured, but there is no control or explanation as to what the gene expression is "relative" to, as depicted in the Y axis. Since these are measurements of genetically modified lines, I would like to see the expression of these genes relative to an isogenic control.

a. Along similar lines, Figure 1E contains highly variable data in the A/G mutants, which is not addressed.

3) The RNA-seq presented in 1K-L shows a lot of altered pancreas development and beta cell pathways in the GO analysis. This is discussed as being interesting, but to me it is expected – the cells being examined are actively being differentiated into pancreas/beta-like cells. It would be more shocking if non-pancreas terms showed up. To me, this interpretation is underwhelming.

4) Figure 2 needs to be explicit about which stage is examined, especially since NKX6.1 somehow returns to regular levels of expression. Are we looking at 1 day after PP1? 2 days? 5 days? It's not clear to me. Since the authors are looking at INS and GCG expression (in 2A, for example), and are normalizing A/A alleles to 1, we don't know how efficient the differentiations normally are. i.e. they show 6x higher GCG expression in the G/G alleles in 2A, but what are the actual values (not as %'s). Could the G/G or A/G alleles be causing cell death at any stage? I would've liked to see these measurements in the study.

5) The identification of RXRa binding site is quite interesting, but there are several issues in Figure 4 that describes this that don't completely make sense to me. Why were PANC1 cells used? These are pancreatic cancer cells, not usually used for studies of islet development. Furthermore, different cell lines (PANC1, HEK293T, Huh7, and hPSC differentiations at various stages) are used in almost every single panel as well – this inconsistency makes the take home point weaker without further explanation as to why the different cell lines were used.

6) In 4E, it appears that RXRa can still occupy the A/A allele. It is stronger in G/G as the authors show, but they also state that an RXRa site is created in the SNP, so how can it still bind to A/A?

7) There are no dose response curves or proper examination of either drug used to inhibit RXRa in this study. Quantification of cell death is also needed with the use of these drugs since RXRs can work in multiple pathways (RA signaling for example) that have broad roles in development.

a. In addition, are these drugs specific to RXRa? What about RXRb or g? These are not adequately discussed.

8) Creating a mouse line is extremely admirable and provides a great potential tool that could address the importance of this SNP in vivo. It is surprising, and exciting, that this point mutation leads to a change in fasted blood glucose levels/GTT

results. Often times compensatory mechanisms will correct for such a small mutation in a non-coding region. This indicates something is special about this SNP. However, figure 5 leaves me with more questions than answers. Understandably, the authors examined islet function and INS/GCG expression using adults. However, the first 4 figures are all examining pancreas development- definitive endoderm/pancreatic progenitor phases that all happen during mouse embryogenesis. These stages of mouse development should be examined here too. Are there more polyhormonal cells during mouse islet differentiation (say at e16.5-e18.5)? Or are there changes in gene expression at these stages that correlate to the findings in the hPSC differentiations?

9) There are again missing controls in Figure 5 upon STZ and HFD challenges. In 5G-I, how do STZ treated A/A, A/G, or G/G animals compare to non-STZ treated? Same question for 5M-O with HFD.

10) If all of the controls were completed, and the authors have identified a functional defect in mice, is there also a functional defect in the human cells? They completed a lot of molecular analyses, but did not look at how this SNP impacted beta-like cell function in their hPSC system, something like GSIS in beta-like clusters for example. Something to test how well human mutant beta-like cells work compared to WT/isogenic controls.

11) Much of the discussion seems to be a rehashing of the introduction including statements that are not fully supported by the data. For instance, in 322-324 the authors claimed to show “functional importance and diabetes susceptibility through human PSC and mouse models”. There were no functional experiments in human models and I would not characterize a slight glucose intolerance in mice susceptibility to diabetes. The STZ and HFD experiments are hard to define given the lack of controls mentioned in 9 above, but my guess is that those treatments blow the SNP out of the water in terms of functional significance.

12) Also in the discussion, the impact of their findings aren't appropriately applied to the real world. In lines 356-360, the authors state “Based on our data that the impaired beta cell differentiation from human PSCs with allele G and higher sensitivity to high-fat-diet and STZ in mice with minor allele G, it reminds that people carrying rs6048205-G should pay extra attention to healthy diet to prevent progressive diabetes especially in developing countries.” Usually people in developing countries do not have the resources to significantly alter their diets, nor would they ever likely or realistically know that they have a SNP in a non coding region of their genomes. Furthermore, how would people with this SNP changing their diets help resist complication from diabetes any more than a person without this SNP? There were no experiments about diet in this study, making this suggestion even less helpful and/or relevant.

Version 1:

Reviewer comments:

Reviewer #1

(Remarks to the Author)

My concerns have been addressed.

Responses to Reviewers' comments

We appreciate the Reviewers' constructive comments on our manuscript. We have accordingly performed substantial experiments to address these comments.

The new data/results include: 1) constructing different cell lines with specific allele A or G which validate that the SNP-G modulates *NKX6-1* expression via activating *FOXA2* at the PP2 stage; 2) control groups added to STZ and HFD challenging experiments, confirming the original conclusions; 3) analyzing the proportion of polyhormonal cells co-expressing SST and INS at the β -like stage, which significantly increases in homozygous G/G cells, with notable more SST and GCG single-positive cells as well; 4) that the SNP locates in an open chromatin region with H3K27ac and H3K4me1 modifications at the PP stage, suggesting it is within an enhancer-like regulatory region; 5) that RXR agonists increased *FOXA2* expression and suppressed *NKX6-1* at the PP2 stage in a dose-dependent manner, without affecting apoptosis levels, while RXR inhibitors repressed *FOXA2* and upregulated *NKX6-1* in a dose-dependent manner, specifically in cells containing SNP-G; 6) immunostaining of the embryonic pancreas showing that G mutation reduces the proportion of NKX6-1+ cells at E13.5 and increases the proportion of polyhormonal cells at E18.5; 7) karyotype analysis confirming the integrity of these cell lines, and genomic PCR assay supporting no detectable on-target or off-target effects in the genome-edited knock-in cell lines; 8) Annexin V/PI assay showing that G variant does not obviously affect apoptosis levels during differentiation.

With regard to the specific comments, our point-by-point responses are listed below:

Reviewer #1:

This manuscript studies pancreatic development and the impact of a pathogenic SNP in this process. The authors use the human pluripotent stem cell model to examine a SNP downstream of the *FOXA2* gene, generating lines that are heterozygous or homozygous for the SNP. They find the SNP leads to abnormal pancreatic development with decreased pancreatic progenitors and skewed differentiation to a polyhormonal endocrine phenotype. Overexpression studies of *FOXA2* also blocks pancreas development and a mouse model of the SNP has defects in glucose homeostasis. Overall, the studies presented are very interesting but there are some issues that need to be addressed. See below for specific comments.

Response: We thank Reviewer 1 for his/her explicit and focused feedback on our manuscript. We hope the revision outlined below will further strengthen our manuscript.

1) The authors only examine a single clone for the human stem cell genome editing. They should show multiple clones to confirm findings are consistent or even better the editing in a second genetically distinct stem cell line would show the effect is

independent of genetic background. This is especially important as the differentiation efficiency shown in figure 1 is very low, with most publications in the field demonstrate at least 30-75% NKX6.1+ cells at the PP2 stage.

Response: As the reviewer suggested, we attempted to edit the ESC line HUES8 available in our lab. However, when we checked the genotype, we found HUES8 is heterozygous in this locus rs6048205 (i.e., A/G). After editing, we failed to get the A/A and G/G clones, but we obtained clones with single allele deletion, described as A/- and G/- (the genotyping results are shown below). With these A/- and G/- cell lines, we performed the *in vitro* differentiation, and the results were consistent with those of homozygous cells. Notably, the G/- genotype did not affect the expression levels of key transcription factors during the definitive endoderm (DE) stage. Consistently, FOXA2 expression showed no significant difference in the DE stage; however, the G variant led to increased FOXA2 expression during the pancreatic progenitor stage (PP1 and PP2). Moreover, G/- did not affect PDX1 expression levels or the percentage of PDX1-positive cells in the PP stage, but it significantly reduced both the percentage and expression levels of NKX6-1-positive cells. These findings support our previous results, reinforcing the robustness of our conclusions regarding the defects in pancreatic progenitor cell differentiation caused by the risk variant G (shown below, also in revised Supplementary Fig. 3).

We realized that the flow cytometry results shown previously were much lower than the immunofluorescence results. Therefore, we have optimized the flow cytometry procedure by adjusting the cell digestion, antibody concentration, and incubation time. Eventually, we detected a 40% positivity rate for PDX1/NKX6-1 at the PP2 stage in A/A cells, which aligns with our immunofluorescence statistical findings (shown below, also in revised Fig. 1i, j).

In addition, edited lines should be tested for genome integrity such as karyotype or CNV analysis.

Response: We thank the reviewer for this suggestion. We have performed karyotype analysis to assess the genome integrity of our edited cell lines. The results indicate that the edited cell lines maintain a normal karyotype (shown below, also in revised Supplementary Fig. 1d).

My worry with CRISPR based homozygous gene editing when analysis is based upon sequencing alone is on target large indels/insertions or rearrangements, see PMID: 35276091. Digital droplet PCR or other methods such as demonstrating another heterozygous SNP is present within the PCR amplicon where the homozygous mutation is introduced to show the homozygous G/G clones have biallelic amplification would be helpful.

Response: We thank the reviewer for reminding us of the concern about on-target. Following the reference ¹, we conducted genomic qPCR to determine the copy number of alleles at the SNP site and observed no obvious differences, indicating there were no on-target large indels/insertions (shown below, also in revised Supplementary Fig. 1e). Additionally, we used online tools to predict potential off-target sites (CCTop - CRISPR/Cas9 target online predictor) and subsequently checked these sites in our cell lines. We found no evidence of off-target effects in those cell lines (shown below, also in revised Supplementary Fig. 1c).

2) In figure 2 the examination of polyhormonal cells only looks at *INS* and *GCG*. *SST* is another common hormone which can present in polyhormonal cells. The analysis should include *SST* cells as well to see if the impact of the variant is only on *GCG* expression or polyhormonal cells in general. When examining *GCG/SST* expression the authors should also look at monohormonal *GCG* and *SST* populations (alpha and delta cells) to see if these are impacted as well.

Response: We thank the reviewer for the comments and accordingly examined the expression level of *SST*. Consistent with *GCG*, *SST* expression was also elevated in G/G line (shown below, also in revised Fig. 2a). More importantly, the population of C-PEP and *SST* double-positive polyhormonal cells was also increased. Following the reviewer's suggestion, we further examined the proportion of hormone-exclusively positive cells, and the monohormonal *GCG* and *SST* populations showed a significant increase (shown below, also in revised Supplementary Fig. 4a), indicating that the risk variant G can impair the normal development of β cells, leading to a preference for other endocrinal lineages.

There is a concern that flow cytometry and microscopy show drastic differences in hormone expression levels. For example in 2c the G/G line has close to 40% GCG+ cells by microscopy but in 2f its only 3%. Why such a huge discrepancy? I worry one of the assays is off. Flow cytometry plots with isotype controls should be shown in supplemental data.

Response: We thank the reviewer for raising such an issue and agree that the positivity rate is relatively low in our flow cytometry data while somewhat high in immunostaining analysis. Our flow cytometry might have underestimated the differentiation efficiency; therefore, we performed a similar optimization to PDX1/NKX6-1 flow analysis. Unfortunately, such optimization did not work well for cytoplasmic proteins like GCG or C-PEP, which may be attributed to suboptimal cell digestion caused by high cell density and heterogeneity. In addition, our fluorescence statistics are calculated based on the ratio of the positive area to the DAPI area. During the later stages of differentiation, the cells grew densely and formed multiple layers, which likely led to an overestimation of differentiation efficiency in our immunostaining analysis, particularly for the cytoplasm-located markers such as C-PEP. Despite the overall lower or higher positive percentage in flow cytometry or immunostaining, respectively, both results were consistent with our RNA, indicating that the G variant increases the proportion of polyhormonal cells and decreases the number of C-PEP single-positive β cells. We have shown the flow cytometry plots with isotype controls below, as seen in the revised Supplementary Figure 4b as well.

3) In Figure 3 the authors do not look at FOXA2 at the protein level in the G/G compared to A/A lines. This would be useful to show in addition to RNA.

Response: We thank the reviewer for this reminding. We performed the Western Blot, and the result was consistent with the RNA data, showing that the G variant leads to a significant increase in FOXA2 protein level at PP2 stage (shown right, also in revised Fig. 3c).

4) In figure 4 agonists of RXR are given and show FOXA2 expression is increased with decreased differentiation efficiency. RA is present in these differentiations which would activate RXR/RAR as well. Can lowering dose of RA correct the defect in the G/G line or at least drop FOXA2 levels? While not absolutely necessary, rescue experiments are more convincing than small molecule experiments that inhibit differentiation because off targets effects of small molecules are a concern.

Response: We thank the reviewer for raising whether RAR plays a role on the G allele during the pancreatic progenitor stage. Xu et al reported that RA inhibited the differentiation of FOXA2-positive floor plate cells and promoted neural differentiation², suggesting that RA may negatively contribute to FOXA2 expression. Indeed, in our experiments using a lower TTNPB dosage (an agonist of RAR signaling), we observed decreased FOXA2 expression. However, NKX6-1 expression was not rescued, likely because lowering RA levels resulted in significantly reduced pancreatic lineage differentiation efficiency, proved by the decreased PDX1 and PTF1A expression (shown below). This suggests that the impact of RA on pancreatic differentiation may be more significant than the effect of the SNP itself.

Regarding the potential off-target effects of the small molecules used in our study, we utilized different chemicals to minimize the off-target effects. The compounds chosen are known for their high specificity for RXR at the given concentrations, as supported by existing literature³⁻⁷.

In addition, as the reviewer suggested, we performed the rescue experiment showing that the RXR inhibitor UVI3003 (UVI) at a low concentration was able to rescue the observed effects: upon UVI treatment, FOXA2 expression decreased in a dose-

dependent manner, while RXR inhibition increased *NKX6-1* expression in G/G cells but not significantly in A/A cells (shown below, also in revised Fig. 4k, Supplementary Fig. 6h, i).

5) Its odd that at the PP2 stage the G/G line has a pretty severe decrease in pancreatic progenitors yet makes more INS+ cells at end stage culture though they are skewed to the polyhormonal fraction? This should at least be discussed in the manuscript.

Response: We thank the reviewer for raising this issue, and we have accordingly added a detailed discussion on this aspect in the revised manuscript (shown on page 14, line 412).

Despite defects in PDX1+/NKX6-1+ pancreatic progenitors, the G variant increased INS-positive cells in the following stage. However, the majority of these INS-positive cells were polyhormonal co-expressed with GCG or SST, and the INS-exclusively positive beta cell was indeed reduced, suggesting that the risk variant G can impair the development of bona fide β cells, leading to a preference for other endocrinal lineages. Although NKX6-1 expression was reduced during the PP stage in G variant cells, it eventually recovered in the late β -like stage, suggesting a delayed expression pattern. Delayed expression of NKX6-1 was reported to result in the formation of polyhormonal cells⁸, which is consistent with our findings that the G mutation leads to an increase in polyhormonal cells. Although NKX6-1 can directly bind to and regulate β -cell-specific transcription factors and insulin processing genes in β -cells⁹, the restricted temporal window for its expression limits its ability to fully correct these developmental defects to prevent the polyhormonal cell fate. Besides, previous studies have reported that while the loss of Nkx6-1 significantly reduces the number of β cells in mouse islets, a small number of β cells are still produced^{10,11}, indicating that other factors may also contribute to the generation of INS-expressing cells. Upon revisiting our SNP PP2 RNA-seq data, we observed that while *NKX6-1*, *PTF1A*, and *SOX9* were downregulated

in the G variant pancreatic progenitors, critical pancreatic transcription factors such as *NKX6-2* and *MNX1* were upregulated. *NKX6-2* was reported to compensate *NKX6-1* in regulating β -cell fate^{11,12}, and *MNX1* was essential for pancreatic β -cell development and identity maintenance^{13,14}. Thus, despite reduced *NKX6-1* expression in pancreatic progenitors with the G variant, the upregulation of these transcription factors may compensate for activating insulin expression.

TPM	A/A-1	A/A-2	G/G-1	G/G-2
NKX6-1	22.94	23.73	5.34	3.95
NKX6-2	5.06	4.31	6.94	8.73
MNX1	21.74	22.65	49.72	51.57
ARX	0.55	0.90	1.85	2.62

Minor issues:

1) Grammar and English language issues throughout the manuscript making it difficult to read. Please correct. An editing service could be helpful here.

Response: We apologize for the issues in writing. We have proofread and corrected grammar and English language errors and believe they have greatly improved.

Reviewer #2 (Remarks to the Author):

The manuscript “A noncoding variant confers pancreatic differentiation defect and contributes to diabetes susceptibility by recruiting RXRa” by Li et al examines the significance of a non coding SNP (rs6048205-G) located downstream of *FOXA2* that is associated with increased blood glucose levels and pancreatic beta cell dysfunction using human pluripotent stem cell (hPSC) and mouse models of islet development and function. Using CRISPR, the authors modeled this SNP in human stem cells as WT(A/A), Het (A/G), and HZ (G/G) alleles, with the G allele being the predicted risk allele. There is some novelty focusing on a single non-coding SNP, although others have examined non coding SNPs in the context of diabetes before (PMID: 32442395, 36778047, 33505025). Here, the authors demonstrate a reduced hPSC derived pancreatic islet differentiation capacity in the presence of the A/G and G/G mutants, with G/G having the most severe impacts and leading to increased formation of polyhormonal cells in the beta-like cell differentiations, specifically working through regulation of *NKX6.1* and *FOXA2*. This analysis includes a couple of valuable RNA-sequencing experiments identifying a broad swath of differentially regulated genes in their G/G mutant. The authors go on to suggest that that this A-G mutation creates an RXRa binding site and complete a series of experiments testing this and trying to determine if RXRa could be responsible for regulating *FOXA2* during pancreatic progenitor specification. In their last figure, the authors also created a new mouse model that contains this single point mutation and showed that adult mice have increased

fasting blood glucose levels and challenging the mice with either STZ depletion of beta cells or a high fat diet, mice carrying the A/G or G/G alleles were more severely impacted compared to A/A controls. While there was quite a bit of effort dedicated to examining this particular SNP and how it impacts the molecular networks that define islet cell differentiation, there are some key controls, experiments, and interpretations of data that are of concern to me:

Response: We thank Reviewer 2 for the summary and comment on our manuscript. We hope the revision outlined below will further strengthen our manuscript.

1) Broadly, is it known that this SNP falls within an enhancer? Are there active enhancers marks that have been shown (H3K27ac or H3K4me1?) Or is this a known open region of chromatin (ATAC-seq)? These datasets are available and should be examined/mentioned here.

a. Similarly, are there any transcripts found in this region (like non-coding RNAs, for example) that could have regulatory implications? How do we know for sure this is a non-coding region?

Response: As the reviewer suggested, we examined the chromatin environment, including accessibility and histone modifications at the locus of SNP rs6048205 based on the published datasets (GSE148368, GSE104840). As shown below, rs6048205 is in an open chromatin region. It exhibits H3K4me1 and H3K27ac modifications in differentiated pancreatic progenitor cells, suggesting that SNP rs6048205 is within an enhancer-like regulatory region (also in revised Supplementary Fig. 6a).

According to the GRCh38 genome annotation, SNP rs6048205 is 321bp upstream of *LINC00261* and 2035bp downstream of *FOXA2*. Additionally, our RNA-seq data show no transcripts observed from this rs6048205 locus (shown below, also in revised Supplementary Fig. 6a).

2) In Figure 1C, the expression of a handful of important beta cell TFs are measured, but there is no control or explanation as to what the gene expression is relative to, as

depicted in the Y axis. Since these are measurements of genetically modified lines, I would like to see the expression of these genes relative to an isogenic control.

Response: We thank the reviewer for pointing this out and apologize for the confusion that arises from how our data are presented. All the RNA expression data from RT-qPCR are calculated relative to the housekeeping gene *GAPDH*. We have now chosen a different type of graph and clearly described it in the figure legends to avoid any misunderstanding (shown below, also in revised Fig. 1c and Fig. 4f).

a. Along similar lines, Figure 1E contains highly variable data in the A/G mutants, which is not addressed.

Response: We thank the reviewer for pointing out the variability in the data presented in Figure 1E for the A/G variants. We repeated the experiment and normalized the data to A/A cells. We have now updated the data (shown below and also in the revised Fig. 1e).

3) The RNA-seq presented in 1K-L shows a lot of altered pancreas development and beta cell pathways in the GO analysis. This is discussed as being interesting, but to me it is expected the cells being examined are actively being differentiated into pancreas/beta-like cells. It would be more shocking if non-pancreas terms showed up. To me, this interpretation is underwhelming.

Response: We are sorry for the statement. Our RNA-seq results supported our RT-qPCR findings, and we observed that the RNA-seq data were consistent with other measurements. We have now revised the description of these results in the manuscript accordingly (shown on page 6, line 165).

4) Figure 2 needs to be explicit about which stage is examined, especially since *NKX6.1* somehow returns to regular levels of expression. Are we looking at 1 day after PP1? 2 days? 5 days? It's not clear to me. Since the authors are looking at *INS* and *GCG* expression (in 2A, for example), and are normalizing A/A alleles to 1, we don't know how efficient the differentiations normally are. i.e. they show 6x higher *GCG* expression in the G/G alleles in 2A, but what are the actual values (not as %s). Could the G/G or A/G alleles be causing cell death at any stage? I would've liked to see these measurements in the study.

Response: We apologize for not clearly describing the stages we tested. All stages presented in Figure 2 are at the β -like cells stage. We have added clarifications in the manuscript and figure legends (shown on page 33, line 955).

The reviewer raised an important question regarding the timing of the increase in *NKX6-1* expression. To address this, we measured *NKX6-1* levels daily following the PP2 stage. After the PP2 stage, we observed that the difference in *NKX6-1* expression levels between the A/A and G/G cells gradually decreased, reaching similar levels by day 5. Interestingly, by day 7 (immature β cell stage), *NKX6-1* expression in the G/G cells had surpassed that in the A/A cells. However, following the PP2 stage, *FOXA2* expression in the G/G cells gradually decreased, reaching levels comparable to those in A/A cells by day 3, and this similarity persisted in the subsequent days, which means that the increased *NKX6-1* expression by day 7 in the G/G cells is independent of *FOXA2* expression levels.

We agree with the reviewer that characterizing differentiation efficiencies is essential. However, rather than presenting the absolute values, we are more interested in highlighting the differences between different genotypes. The actual values from each batch are presented below, showing that the *GCG* expression was consistently higher in G/G cells than in A/A cells at the beta-like stage. Although there were some variations in expression levels among batches, the expression trend in each batch was consistent and aligned with our conclusion. Additionally, our immunofluorescence and flow cytometry data also support our conclusion.

Regarding cell death, we performed the Annexin V/PI analysis for apoptosis in A/A, A/G, and G/G lines at DE, PP1, PP2, and β -like stages. We found that the overall apoptosis rate was very low (<2%), and no apparent changes in apoptosis were observed in the G allele variant cells compared to controls at DE, PP1, or PP2 stages. The results have been included in the revised Supplementary Figure 4d-g.

5) The identification of RXRa binding site is quite interesting, but there are several issues in Figure 4 that describes this that don't completely make sense to me. Why were PANC1 cells used? These are pancreatic cancer cells, not usually used for studies of islet development. Furthermore, different cell lines (PANC1, HEK293T, Huh7, and hPSC differentiations at various stages) are used in almost every single panel as well, this inconsistency makes the take home point weaker without further explanation as to why the different cell lines were used.

Response: We thank the reviewer for raising this issue and apologize for not clearly describing our use of cell lines. PANC1 was clonally derived from human pancreatic duct carcinoma but was also used as an undifferentiated pancreatic progenitor cell¹⁵⁻¹⁷. We used PANC1 to investigate the regulatory function of rs6048205 within the pancreatic lineage context and the hESC-derived pancreatic progenitors. Due to our differentiation findings showing that the SNP does not affect DE and PP1 formation, we hypothesize that the SNP might not play a role in the early stages of differentiation. Therefore, we utilized hPSC differentiation at various stages to demonstrate that the SNP has a stage-specific function.

To validate RXRA binding to the rs6048205-G and enhance the regulatory activity of the G allele sequence, we utilized 293T cells with the concern of minimizing the potential influence of internal regulatory elements specific to the cell line on the SNP sequence and the feasible and simple transfection manipulations in 293T cells. Additionally, to confirm the binding capacity and specificity of RXRA to the SNP sequence using allele-specific ChIP-qPCR, it was crucial to select a heterozygous genetic background with certain *FOXA2* expression. Therefore, we tested several endodermal lineage cell lines and found Huh7 is heterozygous in the rs6048205 locus (so we can directly compare the binding preference on A or G). We have described this more clearly in the revised manuscript.

6) In 4E, it appears that RXRa can still occupy the A/A allele. It is stronger in G/G as the authors show, but they also state that an RXRa site is created in the SNP, so how can it still bind to A/A?

Response: We apologize for the description of the result. Due to the slightly higher background in ChIP-qPCR results, our EMSA results more clearly indicate that RXRA has a much stronger affinity for the risk allele G than the non-risk allele A. We have revised the manuscript to describe this as 'enhanced binding' rather than 'created new RXRA binding site' to more accurately reflect these findings.

7) There are no dose response curves or proper examination of either drug used to inhibit RXRa in this study. Quantification of cell death is also needed with the use of

these drugs since RXRs can work in multiple pathways (RA signaling for example) that have broad roles in development.

Response: We have accordingly provided the dose analysis of RXR activation. We observed that adding an RXR agonist in pancreatic progenitor cells dose-dependently activated RXR target genes. Furthermore, the RNA levels of *FOXA2* increased with higher concentrations of the RXR agonists, while *NKX6-1* expression decreased. Notably, the RXR agonists did not affect *PDX1* expression levels. We have updated the manuscript to include these observations (revised Fig. 4h, Supplementary Fig. 6e, f).

Additionally, we did not observe significant differences in cell death when adding the RXR small molecule during pancreatic differentiation. Using Annexin V/PI staining, we assessed apoptosis levels in pancreatic progenitor cells treated with the highest concentration of the RXR agonists. We found no significant changes (shown below, also in revised Supplementary Fig. 5g, h).

Following the reviewer's suggestion, we treated cells with RXR inhibitor (UVI3003, UVI) during pancreatic differentiation. The results were consistent with those observed using RXR activators. In the G/G cells, the RXR inhibitor did not affect *PDX1* expression but reduced *FOXA2* levels and a slight increase in *NKX6-1* expression. Conversely, in A/A cells, *FOXA2* expression was not reduced, and *NKX6-1* expression

showed no significant change (shown below, also in revised Fig. 4k, Supplementary Fig. 6h, i). Together with our EMSA assay, these findings further support the involvement of RXR regulation through SNP-G interaction.

a. In addition, are these drugs specific to RXRa? What about RXRb or g? These are not adequately discussed.

Response: We thank the reviewer for the helpful discussion. Bexarotene can bind to all three RXR isoforms, exerting its signal transduction ability^{3,18,19} as well as LG268. Due to the structural similarities among RXR subtypes, designing selective agonists for specific RXR subtypes is challenging. No small molecule specifically targets RXRA without affecting RXRB or RXRG^{20,21}. We have revised our descriptions of these small molecules to refer to them as RXR agonists or RXR inhibitors and have added a discussion on RXR subtypes in the revised manuscript (shown on page 15, line 445).

8) Creating a mouse line is extremely admirable and provides a great potential tool that could address the importance of this SNP in vivo. It is surprising, and exciting, that this point mutation leads to a change in fasted blood glucose levels/GTT results. Often times compensatory mechanisms will correct for such a small mutation in a non-coding region. This indicates something is special about this SNP. However, figure 5 leaves me with more questions than answers. Understandably, the authors examined islet function and INS/GCG expression using adults. However, the first 4 figures are all examining pancreas development- definitive endoderm/pancreatic progenitor phases that all happen during mouse embryogenesis. These stages of mouse development should be examined here too. Are there more polyhormonal cells during mouse islet differentiation (say at e16.5-e18.5)? Or are there changes in gene expression at these stages that correlate to the findings in the hPSC differentiations?

Response: We greatly thank the reviewer for appreciating our mouse modeling. Since rs6048205 was annotated with fasting glucose trait, we use adult mice to determine the blood glucose homeostasis. Our results show that the risk variant leads to elevated fasting blood glucose and mild glucose intolerance in adult mice, prompting us to examine pancreatic tissue structure.

Following the reviewer's suggestion, we analyzed the endocrinal cells in the pancreas of E18.5 mice. We observed a low proportion of polyhormonal cells in the embryonic mouse pancreas. However, quantification of these polyhormonal cells in the G/G group revealed a significant increase. Additionally, NKX6-1 positive cells were markedly reduced in the E13.5 embryonic pancreas, consistent with the phenotype observed in our *in vitro* differentiation (shown below, also in revised Fig. 5f, g Supplementary Fig. 7e, f).

9) There are again missing controls in Figure 5 upon STZ and HFD challenges. In 5G-I, how do STZ treated A/A, A/G, or G/G animals compare to non-STZ treated? Same question for 5M-O with HFD.

Response: Following the reviewer's suggestion, we have added the relevant control experiments. Our conclusions remain consistent with the previous results. Detailed descriptions are as follows:

For STZ challenges, we conducted additional experiments using 6-8 week-old mice of different genotypes from the same litter. The mice were randomly divided into two groups, receiving either low-dose STZ or citrate buffer (the solvent for STZ). We observed that A/A and heterozygous A/G mice showed a slight increase in random blood glucose levels compared to the solvent-injected group. In contrast, homozygous G/G mice had the highest increase in random blood glucose levels. Subsequently, we performed GTT assay on these mice. We found low-dose STZ induced more pronounced glucose intolerance in mice, with homozygous G/G mice showing the most significant glucose intolerance (shown below, also in revised Fig. 5i, j, Supplementary Fig. 7g).

Due to the extended duration of high-fat diet (HFD) feeding in previous experiments, we were unable to conduct a new set of experiments with mice from the same litter within a short timeframe. However, we re-evaluated blood glucose levels in normally fed mice of the same age. We found that random blood glucose levels in the mice with a chow diet did not increase significantly. Following HFD, all genotypes exhibited a notable increase in glucose intolerance and fasting blood glucose levels compared to the normally fed group, with homozygous mutant mice showing the highest levels of glucose intolerance and fasting blood glucose (shown below, also in revised Fig. 5m, n, Supplementary Fig. 7h).

10) If all of the controls were completed, and the authors have identified a functional defect in mice, is there also a functional defect in the human cells? They completed a lot of molecular analyses, but did not look at how this SNP impacted beta-like cell function in their hPSC system, something like GSIS in beta-like clusters for example. Something to test how well human mutant beta-like cells work compared to WT/isogenic controls.

Response: We thank the reviewer for this important suggestion. To address whether the functional defect observed in mice is also present in human cells, we conducted GSIS (Glucose-Stimulated Insulin Secretion) assays on differentiated human cells. The existence of Insulin in the media (supplemented with ITS-X), used for the beta-like stage, might affect the assay of Insulin secretion; therefore, we also provide C-PEP measurement results. Our results show that the G/G cells exhibit reduced Insulin and C-PEP secretion in response to high glucose stimulation. This suggests that the functional defect is also evident in human cells. We have included these findings in the revised manuscript (shown below, and also in revised Fig. 2g, h).

11) Much of the discussion seems to be a rehashing of the introduction including statements that are not fully supported by the data. For instance, in 322-324 the authors claimed to show 'functional importance and diabetes susceptibility through human PSC and mouse models'. There were no functional experiments in human models and I would not characterize a slight glucose intolerance in mice susceptibility to diabetes. The STZ and HFD experiments are hard to define given the lack of controls mentioned in 9 above, but my guess is that those treatments blow the SNP out of the water in terms of functional significance.

Response: We thank the reviewer for highlighting our discussion section's shortcomings. We have reorganized the discussion in the revised manuscript and provided a more objective description of our experimental results.

Under normal conditions, G/G mice displayed mildly elevated fasting blood glucose and slight glucose intolerance. However, following low-dose streptozotocin (STZ) or a high-fat diet (HFD), their blood glucose levels significantly increased, with some reaching diabetic thresholds. After 40 days of low-dose STZ treatment, 3/4 of G/G mice had random blood glucose levels exceeding 16.7 mmol/L, while 2/3 showed fasting blood glucose above 11.1 mmol/L after HFD. Both conditions led to severe glucose intolerance, indicating most G/G mice were prone to develop diabetic symptoms upon stimulation. In contrast, while WT mice exhibited slight blood glucose increases under the same stimuli, their levels remained within normal ranges, highlighting the G/G

mice's susceptibility to diabetes. These results suggest that the SNP-G increases the basal blood glucose level, thus increasing diabetes susceptibility in mice.

12) Also in the discussion, the impact of their findings aren't appropriately applied to the real world. In lines 356-360, the authors state 'Based on our data that the impaired beta cell differentiation from human PSCs with allele G and higher sensitivity to high-fat-diet and STZ in mice with minor allele G, it reminds that people carrying rs6048205-G should pay extra attention to healthy diet to prevent progressive diabetes especially in developing countries.' Usually people in developing countries do not have the resources to significantly alter their diets, nor would they ever likely or realistically know that they have a SNP in a non-coding region of their genomes. Furthermore, how would people with this SNP changing their diets help resist complication from diabetes any more than a person without this SNP? There were no experiments about diet in this study, making this suggestion even less helpful and/or relevant.

Response: We thank the reviewer for highlighting the inappropriate discussion on diet recommendations for people in developing countries. We have removed these sentences from the discussion in the revised manuscript.

References:

- 1 Simkin, D. *et al.* Homozygous might be hemizygous: CRISPR/Cas9 editing in iPSCs results in detrimental on-target defects that escape standard quality controls. *Stem Cell Reports* **17**, 993-1008, doi:10.1016/j.stemcr.2022.02.008 (2022).
- 2 Xu, T. *et al.* Uncovering the role of FOXA2 in the Development of Human Serotonin Neurons. *Adv Sci (Weinh)* **10**, e2303884, doi:10.1002/advs.202303884 (2023).
- 3 Liu, Y. *et al.* The novel function of bexarotene for neurological diseases. *Ageing Res Rev* **90**, 102021, doi:10.1016/j.arr.2023.102021 (2023).
- 4 Ma, X. *et al.* Retinoid X receptor alpha is a spatiotemporally predominant therapeutic target for anthracycline-induced cardiotoxicity. *Science advances* **6**, eaay2939, doi:10.1126/sciadv.aay2939 (2020).
- 5 Rambow, F. *et al.* Toward Minimal Residual Disease-Directed Therapy in Melanoma. *Cell* **174**, 843-855 e819, doi:10.1016/j.cell.2018.06.025 (2018).
- 6 Cao, X. *et al.* Chromatin accessibility dynamics dictate renal tubular epithelial cell response to injury. *Nat Commun* **13**, 7322, doi:10.1038/s41467-022-34854-w (2022).
- 7 Perez, E., Bourguet, W., Gronemeyer, H. & de Lera, A. R. Modulation of RXR function through ligand design. *Biochim Biophys Acta* **1821**, 57-69, doi:10.1016/j.bbap.2011.04.003 (2012).
- 8 Russ, H. A. *et al.* Controlled induction of human pancreatic progenitors produces functional beta-like cells in vitro. *EMBO J* **34**, 1759-1772, doi:10.15252/emj.201591058 (2015).
- 9 Taylor, B. L., Liu, F. F. & Sander, M. Nkx6.1 is essential for maintaining the functional state of pancreatic beta cells. *Cell Rep* **4**, 1262-1275, doi:10.1016/j.celrep.2013.08.010

- (2013).
- 10 Sander, M. *et al.* Homeobox gene Nkx6.1 lies downstream of Nkx2.2 in the major pathway of beta-cell formation in the pancreas. *Development* **127**, 5533-5540, doi:10.1242/dev.127.24.5533 (2000).
- 11 Nelson, S. B., Schaffer, A. E. & Sander, M. The transcription factors Nkx6.1 and Nkx6.2 possess equivalent activities in promoting beta-cell fate specification in Pdx1+ pancreatic progenitor cells. *Development* **134**, 2491-2500, doi:10.1242/dev.002691 (2007).
- 12 Schaffer, A. E., Freude, K. K., Nelson, S. B. & Sander, M. Nkx6 transcription factors and Ptf1a function as antagonistic lineage determinants in multipotent pancreatic progenitors. *Dev Cell* **18**, 1022-1029, doi:10.1016/j.devcel.2010.05.015 (2010).
- 13 Dalgin, G. *et al.* Zebrafish mnx1 controls cell fate choice in the developing endocrine pancreas. *Development* **138**, 4597-4608, doi:10.1242/dev.067736 (2011).
- 14 Wang, H., Wei, X., Shi, W., He, J. & Luo, L. Key Developmental Regulators Suggest Multiple Origins of Pancreatic Beta Cell Regeneration. *Zebrafish*, doi:10.1089/zeb.2019.1777 (2020).
- 15 Hardikar, A. A., Marcus-Samuels, B., Geras-Raaka, E., Raaka, B. M. & Gershengorn, M. C. Human pancreatic precursor cells secrete FGF2 to stimulate clustering into hormone-expressing islet-like cell aggregates. *Proc Natl Acad Sci U S A* **100**, 7117-7122, doi:10.1073/pnas.1232230100 (2003).
- 16 Wu, Y. *et al.* c-Kit and stem cell factor regulate PANC-1 cell differentiation into insulin- and glucagon-producing cells. *Lab Invest* **90**, 1373-1384, doi:10.1038/labinvest.2010.106 (2010).
- 17 Dadheech, N., Srivastava, A., Shah, R. G., Shah, G. M. & Gupta, S. Role of poly(ADP-ribose) polymerase-1 in regulating human islet cell differentiation. *Sci Rep* **12**, 21496, doi:10.1038/s41598-022-25405-w (2022).
- 18 Vu-Dac, N. *et al.* Retinoids increase human apolipoprotein A-11 expression through activation of the retinoid X receptor but not the retinoic acid receptor. *Mol Cell Biol* **16**, 3350-3360, doi:10.1128/MCB.16.7.3350 (1996).
- 19 Chitranshi, N., Dheer, Y., Kumar, S., Graham, S. L. & Gupta, V. Molecular docking, dynamics, and pharmacology studies on bexarotene as an agonist of ligand-activated transcription factors, retinoid X receptors. *J Cell Biochem* **120**, 11745-11760, doi:10.1002/jcb.28455 (2019).
- 20 de Almeida, N. R. & Conda-Sheridan, M. A review of the molecular design and biological activities of RXR agonists. *Med Res Rev* **39**, 1372-1397, doi:10.1002/med.21578 (2019).
- 21 Leal, A. S., Hung, P. Y., Chowdhury, A. S. & Liby, K. T. Retinoid X Receptor agonists as selective modulators of the immune system for the treatment of cancer. *Pharmacol Ther* **252**, 108561, doi:10.1016/j.pharmthera.2023.108561 (2023).